# Efficient Uncertainty Estimation via Sensitivity-Guided Subnetwork Selection for Scalable Variational Inference

## Abstract

Quantifying predictive uncertainty with minimal computational overhead remains a significant challenge for reliable deep learning applications in safety-critical systems. While Bayesian neural networks (BNNs) are the gold standard for uncertainty quantification, they require considerable training time and computational resources. Although a body of work has focused on reducing the computational cost of BNN inference through post-hoc approaches, fewer efforts have addressed accelerating training and improving scalability for large-scale image applications. This paper proposes a partial Bayesian training approach via mean-field variational inference (VI), enabling controllable uncertainty modeling through sparse gradient representations. The selection of the variational Bayesian subnetwork is guided by sensitivity analysis using a first-order approximation, which is grounded in uncertainty propagation theory. Under mean-field assumptions, we demonstrate how this framework effectively informs the selection of parameters that represent the network's predictive uncertainty. This criterion is efficiently supported by standard auto-differentiation tools and is available out of the box, requiring no additional implementation or computational overhead. The resulting model consists of a combination of deterministic and Bayesian parameters, facilitating an effective, yet efficient, representation of uncertainty. We investigate the effects of varying the proportion of Bayesian parameters (ranging from 1% to 95%) across diverse tasks, including regression, classification, and semantic segmentation. Experimental results in MNIST, CIFAR-10, ImageNet, and Cityscapes demonstrate that our approach achieves competitive performance and uncertainty estimates compared to ensemble methods. While maintaining substantially fewer parameters, approximately 45%, 80% less than standard VI and ensembles, this approach offers a scalable solution with reduced training costs compared to standard VI or partial VI trained from random initialization. Furthermore, we assess the robustness of predictive uncertainty in the presence of covariate shifts and out-of-distribution data, demonstrating that this method effectively captures uncertainty and exhibits robustness to image corruptions.

## 1 Introduction

Robust uncertainty quantification is essential to ensure reliable decision-making in safety-critical systems (Nguyen et al., 2015; Guo et al., 2017; Yang et al., 2023; Cao et al., 2024), such as medical diagnosis (Rajpurkar et al., 2022) and self-driving cars (Bojarski et al., 2016). Predictive uncertainty is a quantitative metric for facilitating risk assessment for real-world deployment by providing insights into the model's confidence in its predictions. Risk assessment related to distributional shifts in real-world environments compared to the training i.i.d. dataset is enabled through calibrated predictive uncertainty (Malinin et al., 2021; 2022; Ovadia et al., 2019). Poorly calibrated probabilities are associated with silent failures, where models exhibit overconfident false predictions (Guo et al., 2017). Bayesian neural networks (BNN) (Ghahramani, 2015) provide an approach to learning model uncertainty over parameter distributions. BNNs, however, suffer from a large parameter space, slow convergence, and noisy loss landscapes limited by sampling (Jospin et al., 2022). As a result, applications of BNNs to complex tasks such as large multi-class classification or segmentation remain challenging (Ovadia et al., 2019). Deterministic network pruning, predefined sparsity,

and learning sparse networks have shown success in preserving predictive performance while reducing computational overhead (Evci et al., 2020; Frankle & Carbin, 2019; Mozer & Smolensky, 1988; LeCun et al., 1989). Partial Bayesian Neural Networks are effective at preserving predictive uncertainty over full-BNNs; however, current efforts rely on an iterative selection of the layer-wise subnetwork (Zeng et al., 2018; Sharma et al., 2023), constructing a low-dimensional parameter space through PCA (Izmailov et al., 2020), or on a computationally limiting Hessian-based subnetwork selection (Daxberger et al., 2021b). We demonstrate a computationally low-cost method based on first-order sensitivity analysis to select a sparse subnetwork for partial Bayesian inference to reduce overparameterization in BNNs; our contributions are as follows:

1. Examine an algorithm for training scalable Partial Bayesian Neural Networks (PBN)[1] that selectively assigns a subnetwork of variational Bayesian parameters based on sensitivity analysis;

2. Demonstrate the scalability of the method based on empirical assessments on large-scale classification on ImageNet and semantic segmentation on CityScapes.

3. Evaluate the predictive uncertainty and its reliability on out-of-distribution testing on covariate data shifts and unseen classes.

## 2   Related works

Predictive uncertainty in deep learning can be estimated using a variety of approaches, with Bayesian neural networks (BNNs) being a principled and widely studied class of methods. Unlike standard neural networks that learn point estimates, BNNs model distributions over parameters, enabling the capture of epistemic (model) uncertainty (Abdar et al., 2021). Practical approximations to Bayesian inference include variational inference (VI) (Blundell et al., 2015), Markov Chain Monte Carlo (MCMC) methods (Welling & Teh, 2011), dropout-based Monte Carlo sampling (Gal & Ghahramani, 2016; Kingma et al., 2015), and expectation propagation (Hernández-Lobato & Adams, 2015). Outside the Bayesian paradigm, deep ensembles (Lakshminarayanan et al., 2017) offer a simple yet effective method for uncertainty estimation by aggregating predictions from multiple independently trained models. To reduce the training and memory cost of ensembling, BatchEnsembles (Wen et al., 2020) introduce shared weights and mini-batch parallelization. Despite the wide range of techniques available for predictive uncertainty estimation (Abdar et al., 2021), many suffer from high computational overheads during both training and inference, limiting their scalability to large-scale applications.

**Overparameterization.** Reducing parameter complexity has proven to be an effective strategy for training sparse deterministic neural networks (Guo et al., 2016; Dey et al., 2019; Evci et al., 2020; LeCun et al., 1989). Sparsity can be achieved through methods such as connection pruning (Guo et al., 2016; LeCun et al., 1989), random pre-defined sparsity patterns (Dey et al., 2019), or dynamic drop-and-grow algorithms that adaptively modify the network based on parameter saliency (Evci et al., 2020). Parameter saliency, which measures the importance of individual parameters, is often used to guide pruning decisions (Yeung et al., 2010). This importance is typically assessed via sensitivity analysis, examining how perturbations to parameters affect the output of the loss or objective function. Due to the high computational cost of exact calculations, such analyses commonly rely on first- and second-order approximations (Yeung et al., 2010). Sensitivity-based pruning techniques have been successfully applied in various settings, including in-training pruning (Evci et al., 2020; Shi et al., 2020), post-hoc pruning (Mozer & Smolensky, 1988; Molchanov et al., 2019), and iterative pruning methods such as the Lottery Ticket Hypothesis (Frankle & Carbin, 2019). While these approaches have primarily focused on deterministic neural networks, we next consider parameter reduction strategies within probabilistic (Bayesian) models.

**Computational Demand of BNNs.** Various techniques have been proposed to accelerate inference in Bayesian neural networks (BNNs), often as post-hoc processing steps such as pruning (Sharma & Jennings, 2021) or quantization (Subedar et al., 2021; Ferianc et al., 2021) following the training of a dense BNN (Jia et al., 2021; Subedar et al., 2021). However, these approaches incur additional computational costs and rely

---

[1]Partial Bayesian Neural Networks in this context refers to sparsity introduced into the sigma parameter, rendering the network "partially" Bayesian.

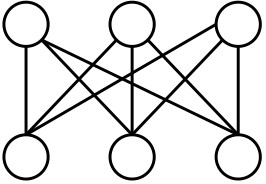
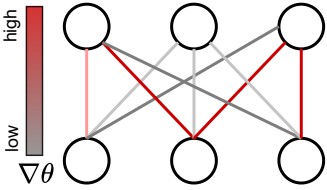
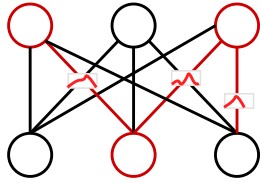

(a) Deterministic Training          (b) Subnetwork Selection          (c) Partial VI Training

Figure 1: Method for training partial variational Bayesian networks (PBNs) consists of three stages: (a) Train a deterministic model by minimizing the negative log-likelihood, with parameters $\theta$ initialized randomly. (b) Perform first-order gradient-based sensitivity analysis to compute $\nabla\theta$ and identify the top-$k$ parameters by magnitude $|\nabla\theta|$, which are then modeled as stochastic variables; the rest remain deterministic. The learned weights from (a) are used to initialize this step. (c) Train the resulting PBN using variational inference by maximizing the Evidence Lower Bound (ELBO).

on first training a full, often expensive, BNN. Moreover, BNNs frequently face convergence challenges and noisy loss landscapes, stemming from the limited number of posterior samples drawn during training (Jospin et al., 2022; Ovadia et al., 2019). While increasing the number of samples theoretically improves the posterior approximation, it is often impractical due to the significant computational overhead involved in sampling at each gradient update (Jospin et al., 2022). To address scalability, the Stochastic Weight Averaging-Gaussian (SWAG) method approximates the posterior using the first moment of SGD iterates combined with a low-rank plus diagonal covariance structure (Maddox et al., 2019). SWAG offers a computationally efficient alternative to ensembles, delivering better-calibrated uncertainty estimates with a single model solution (Maddox et al., 2019).

**Partial Bayesian Neural Networks.** Partial Bayesian learning leverages the computational efficiency of deterministic networks with the probabilistic representation of Bayesian parameters. (Zeng et al., 2018) investigated the placement of Bayesian layers in active learning, showing that using one or two near the output outperforms fully Bayesian models. A more recent study has explored partial Bayesian learning to estimate uncertainty within a brain tumor segmentation model (Prabhudesai et al., 2023). (Daxberger et al., 2021b) utilizes a linearized Laplace approximation to infer a Bayesian subnetwork within a pre-trained deterministic network. At inference, they employ an approximate Hessian with a generalized Gauss-Newton (GGN)-Laplace method to infer a full-covariance Gaussian posterior. Their method is validated on regression and classification tasks on MNIST and CIFAR10 datasets. While achieving competitive uncertainty quantification akin to ensembles, their approach requires the computation and storage of the covariance matrix, rendering it intractable for tasks such as semantic segmentation or large-multi-class classification.

Sharma et al. (2023) evaluated stochastic subset inference on a two-step training approach as (Daxberger et al., 2021b) through Hamiltonian Monte Carlo (HMC) (Neal, 2012), Laplace Approximation, SWAG (Maddox et al., 2019) on regression and one-step joint training using stochastic variational inference (SVI) (Blundell et al., 2015) on classification. They analyzed the impact of sub-stochastic parameter placement, concluding that stochastic input layers yield the highest accuracy, while stochastic last-ResBlock and output layers have the lowest negative log-likelihood. Last layer Laplace approximation has also been proposed as a post-hoc computationally efficient method of Bayesian approximation (Daxberger et al., 2021a). A variational Bayesian last layer approach was demonstrated offering a sampling-free VI (Harrison et al., 2024) proposes a more deterministic formulation for training Bayesian last layers for addressing the sampling issue associated with BNNs; however, the method introduces additional hyperparameters that potentially limit its practical introduction for real-world applications.

Our contribution builds on the method proposed by Abboud et al. (Abboud et al., 2024), which introduced subnetwork inference for efficient Bayesian deep learning. We extend this line of work by formalizing the approach through the lens of uncertainty propagation and demonstrating its scalability to large-scale image

applications. In doing so, we present a computationally efficient method for training partial variational inference models that reduces parameter complexity and scales to datasets such as ImageNet (Deng et al., 2009).

## 3 Method

Given a dataset $\mathcal{D}$, with input samples $\mathbf{x} \in \mathbb{R}^D$ and targets $y$, such that y is real-valued for regression, categorical for classification, and segmentation. We aim to learn a neural network $f_\Theta(\cdot)$, parameterized by $\Theta = \theta_1, \theta_2, ..., \theta_i$ for a given task by minimizing the error $E$, such that $\min_\Theta E(\mathcal{D}, \Theta) = \min_\Theta E(y|x; \Theta)$

To estimate the uncertainty in the output $y$ of a neural network $f_\Theta(\cdot)$, it is essential to account for the uncertainties associated with its parameters. This can be achieved by leveraging the theory of *uncertainty propagation* (Ku et al., 1966; Zurada et al., 1994; Morgan & Henrion, 1990), which quantifies how uncertainty in a model's parameters induces uncertainty in the output. As a prerequisite, uncertainty analysis must be conducted to quantify the individual contributions of parameter-level uncertainties to the total output uncertainty. We assume that the parameters are not all equally important based on existing evidence (Daxberger et al., 2021b; Abboud et al., 2024; Izmailov et al., 2020; Evci et al., 2020). Therefore, to carry out uncertainty analysis, we must first quantify individual-parameter importance through a measure of *uncertainty importance*. A simple measure of uncertainty importance is sensitivity; the rate at which the output responds to changes in the input parameters. The sensitivity is represented by the partial derivative of the output with respect to the individual parameter $\partial f(x)/\partial \theta$.

We now derive the relationship between the uncertainty importance measure, sensitivity, and the resulting uncertainty in the model output. For the network $f$, parameterized by $\Theta$ (for brevity and ease of readability, we will refer to $f_\Theta(x)$ with $f(\Theta)$ as $\Theta$ is the random parameter of interest in this case), the variance in the output is defined as follows,

$$Var(y) = \mathbb{E}\left[(f(\Theta) - \mathbb{E}[f(\Theta)])^2\right] = \mathbb{E}\left[f(\Theta)^2\right] - (\mathbb{E}[f(\Theta)])^2 \tag{1}$$

We can derive approximations[2] for the terms $\mathbb{E}[f(\Theta)^2]$ and $(\mathbb{E}[f(\Theta)])^2$ by performing a Taylor series expansion of $f(\Theta)$ about the mean parameter value $\mu_\Theta$, such that

$$\mathbb{E}[f(\Theta)] \approx f(\mu_\Theta) + \frac{1}{2}f''(\mu_\Theta)\sigma_\Theta^2 \tag{2}$$

$$\mathbb{E}[f(\Theta)^2] \approx f(\mu_\Theta)^2 + \left(f'(\mu_\Theta)^2 + f(\mu_\Theta)f''(\mu_\Theta)\right)\sigma_\Theta^2 \tag{3}$$

Then by substituting equations 2, 3 into equation 1:

$$\therefore Var(y) \approx f'(\mu_\Theta)^2\sigma_\Theta^2 - \frac{1}{4}f''(\mu_\Theta)^2\sigma_\Theta^4$$
$$\approx f'(\mu_\Theta)^2\sigma_\Theta^2 \tag{4}$$
$$\therefore \sigma_y^2 = \left(\frac{\partial f}{\partial \Theta}\right)^2 \sigma_\Theta^2$$

The first term in equation 4 is the first-order Taylor approximation, where $\sigma_y^2$ and $\sigma_\theta^2$ represent the variances in $y$ and $\Theta$, respectively. Since we are employing mean-field theory, we can decompose the right-hand side of equation 4 into the individual network parameters, leveraging the mean-field assumption of parameter independence (Blei et al., 2017; Farquhar et al., 2020):

---

[2]Full derivation in Appendix J

$$\sigma_y^2 \approx \left(\frac{\partial f}{\partial \theta_1}\right)^2 \sigma_{\theta_1}^2 + \left(\frac{\partial f}{\partial \theta_2}\right)^2 \sigma_{\theta_2}^2 + \ldots + \left(\frac{\partial f}{\partial \theta_N}\right)^2 \sigma_{\theta_N}^2 \tag{5}$$

$$\therefore \sigma_y^2 = \sum_{i=1}^{N} \left(\frac{\partial f}{\partial \theta_i}\right)^2 \sigma_{\theta_i}^2 \tag{6}$$

The term $\partial f / \partial \theta_i$ quantifies the sensitivity of the output uncertainty to the uncertainty in individual parameters $\theta_i$, and therefore represents the individual uncertainty importance metric. Equation 6 is therefore the relationship between the output uncertainty, sensitivity, and individual parameter uncertainty that allows us to determine the subnetwork of important uncertainty parameters to represent the network uncertainty.

To implement this, we formulate an algorithm for training Partial Bayesian Neural Networks, as depicted in Figure 1 and Algorithm 1 based on a method described in Abboud et al. (2024). Given a pre-trained deterministic model, a partial Bayesian model is then trained with MFVI with a Gaussian prior, initializing the mean values $\mu$ with the point estimates from the deterministic model. The selection of Bayesian parameters is based on sensitivity analysis using a first-order approximation, where the number of Bayesian parameters is controlled by the hyperparameter $r_{bayes}$, which adjusts the proportion of Bayesian parameters in the final Partial Bayesian Neural Network.

**Notation:** We aim to learn a partial Bayesian neural network $f_{\Theta_b}(\cdot)$, where $\Theta_b$ is parameterized by $(\mu_b, \sigma_b) \in \mathbb{R}^N$, and their respective means $\mu_b$ are initialized from a pre-trained deterministic neural network $f_{\Theta_d}$, where $\Theta_d \in \mathbb{R}^N$. For weights designated as Bayesian, their corresponding $\sigma_b$ values are modeled as the softplus $(\sigma = log(1 + exp(\rho)))$ of a randomly initialized $\rho$ parameter sampled from a Gaussian distribution and minimized by the $KL$ divergence against a standard normal prior $\mathcal{N}_{prior}(0, I)$. The loss is defined by the evidence lower bound criterion (ELBO), $\mathcal{L}_{ELBO} = \sum_i \mathcal{L}(f_{\Theta_d}(x_i), y_i) + \beta \cdot KL(q(\theta_b), p(\theta_b))$, where $q(\theta_b), p(\theta_b)$ are the posterior and prior terms over the parameters, $\beta$ is the weight of the KL divergence term. The KL weight term, $\beta$, is gradually annealed over training epochs, from an initial value $\beta_{init}$ to a target value $\beta_{target}$ according to the schedule $\beta_i = \beta_{target} - (\beta_{target} - \beta_{init}) \times (epoch_i/epoch_{total})$. Subscripts $b$ and $d$ represent Bayesian and deterministic parameters, respectively.

**Training the Deterministic Network:** The first step in our approach is to train a deterministic model on a given dataset $\mathcal{D} = \{x_n, y_n\}_{n=1}^N$ composed of i.i.d. data points. We train a neural network to model the function $f_{\Theta_d}(\cdot)$ parameterized by $\Theta_d$, by minimizing the negative log-likelihood $\mathcal{L}(f_{\Theta_d}(x), y)$ as a standard neural network with point estimates.

**Initialization of Partial Bayesian Neural Network:** Before training the Partial Bayesian Neural Network, a gradient-based sensitivity analysis (Yeung et al., 2010) (Algorithm 1- Step 8) is used to select the weights with the highest magnitude gradients, noted as $Top k(|\nabla_\Theta|)$, where $k$ is the number of parameters to set as Bayesian based on an input hyperparameter $r_{bayes}$. We use first-order gradients to select the parameters with the highest-magnitude gradients and reparameterize them as variational parameters. The benefit

---

**Algorithm 1** Partial Bayes with Sparse Gradients

---

1: **Training Step 1**: Deterministic
2:    **Input:** Dataset $D$. Initialize $\Theta_d$. **Learn** $f_{\Theta_d}$ by minimizing $\mathcal{L}(f_\theta(x_i), y_i)$. **Output**: $f_{\Theta_d}$
3: **Training Step 2**: Partial Bayesian
4:    **Input:** Network $f_{\Theta_d}$, dataset $D$, Bayesian rate $r_{bayes}$, Posterior init parameters $\mu_{post}, \sigma_{post}$
5:    **Initialize**:
6:       $\Theta_b; (\mu_b, \sigma_b) \leftarrow$ Bayes parameters
7:       $\nabla_\Theta \mathcal{L} \leftarrow$ compute gradients
8:       $Top k(|\nabla_\Theta|) \leftarrow$ Sensitivity Analysis ($k = r_{bayes} \times N_{\theta_{total}}$)
9:       $\mu_{b,d} = \theta_d \leftarrow \Theta_d, \quad \sigma_b = log(1 + exp(\rho_b)), \quad \rho_b \sim \mathcal{N}(\mu_{init}, \sigma_{init}), \quad \sigma_d = 0$
10:    **Learn** $f(\Theta_b, \Theta_d)$ by minimizing $\mathcal{L}(f_\theta(x), y) + \beta \cdot KL(q(\theta_b), p(\theta_b))$

---

of using first-order approximations is that it is provided by automatic differentiation tools such as PyTorch at no additional computational cost to the workflow (Paszke et al., 2017).

The Top-$k$ first-order gradient selection method enables a mixture of deterministic and Bayesian parameters within each layer or filter, introducing sparsity into the $\sigma$ parameter. Deterministic weights are modeled as delta functions $\delta(\mu_i)$, while Bayesian weights follow distributions $\mathcal{N}(\mu_i, \sigma_i)$, with $\mu_i$ initialized from the deterministic model's point estimates to provide a robust starting point for uncertainty learning in the Partial Bayesian Neural Network.

For each layer $l$, a mask is constructed based on the gradient magnitudes of the layer's parameters to determine which parameters will be treated as Bayesian. The corresponding $\rho^{[l]}$ values in layer $l$ are initialized as follows: Bayesian parameters $\rho_b$ are sampled from a normal distribution, while deterministic parameters $\rho_d$ are initialized near zero to ensure numerical stability. During training, the $\rho_d$ values are frozen by setting their gradients to zero, ensuring they do not contribute to the KL divergence term. A visual illustration of this selective gradient update process is provided in Appendix I.

**Training the Partial Bayesian Neural Network:** The Partial Bayesian Neural Network (PBN)[3] is trained using variational inference (Kingma & Welling, 2014) by employing the reparameterization trick (Kingma et al., 2015). Five posterior samples were chosen, as experimenting with more samples slightly improved performance but significantly increased training time. A 5-member ensemble was run as a comparative baseline. The network is trained to minimize the -ELBO loss described in Algorithm 1. During this step, both deterministic and Bayesian parameters are jointly updated.

**Sparse Gradient Updates:** The layers of the PBN consist of a combination of deterministic and Bayesian parameters requiring customization of the forward and backward passes. Sparse gradient representations enable partial updates to the $\rho_i$ parameters. The gradient tensor associated with the $\rho$ tensor is then converted into a sparse representation, where $\nabla \rho_d = 0; \forall \theta_d$ instructing the optimizer to disregard these elements within the computational graph. Consequently, the $\rho_d$ elements associated with deterministic weights remain unchanged in the computational graph, as their gradients are zero. This is in contrast with the methods described by (Prabhudesai et al., 2023; Zeng et al., 2018) that require calculating and storing full gradients for each $\rho$ parameter at each optimization step.

## 4 Experimental Results

The following experiments are designed to demonstrate the versatility and scalability of the proposed method across a range of tasks, from simple regression to multi-class classification and complex multi-class segmentation, with network sizes spanning from $10^2$ to $10^7$ parameters. In the following experiments, we address several key questions: (1) Is the uncertainty represented by the Partial Bayesian Network well-calibrated under distributional shifts? (2) Is the proposed method scalable to large networks and datasets? We also examine the distribution of the selected subnetwork parameters within the model and compare it with prior work to understand structural differences. Additionally, we investigate whether initializing the PBN's from deterministic point estimates facilitates training. By exploring these questions, we aim to better understand how the subnetwork method in PBN behaves under realistic deployment conditions, assess their practicality at scale, and identify design choices—such as parameter selection and initialization—that impact their uncertainty quality and training efficiency.

**Datasets:** We evaluated our methods on several benchmark datasets. For classification, we used MNIST (LeCun, 1998), CIFAR-10 (Krizhevsky et al., 2009), and ImageNet (Deng et al., 2009). The segmentation experiments were conducted on the Cityscapes dataset (Cordts et al., 2016). To assess robustness under covariate shift, we used the CIFAR-10-C and ImageNet-C datasets (Hendrycks & Dietterich, 2019) for out-of-distribution (OOD) testing.

**Evaluation metrics:** Accuracy and intersection-over-union (IoU) is used to evaluate classification and segmentation performances, respectively. The Brier score is used as a proper scoring rule for measuring uncertainty (Brier, 1950; Ovadia et al., 2019) for Brier Score $= \frac{1}{N} \sum_{i=1}^{N} (\delta_{i=y} - p_\theta(y = C \mid x))^2$. Negative

---

[3]Partial Bayesian Neural Network (PBN) with $r_{\text{bayes}} = R\%$ is referred to in the paper as PBN R% or Partial R%

log-likelihood (NLL) is also used to evaluate the quality of uncertainty (Ovadia et al., 2019). Entropy of Expectation (EoE) is used to compute the total uncertainty $EoE = -p(y = c \mid x) \log(p(y = c \mid x))$. Floating point operations (FLOPs) and the number of model parameters are used to measure computational efficiency (details in Appendix:C).

## 4.1 Regression on Toy Dataset

We first evaluate our approach qualitatively on a one-dimensional toy regression dataset. We follow the same experimental setup described by (Hernández-Lobato & Adams, 2015). The toy dataset is drawn from $y = x^3 + \epsilon$, where $\epsilon \sim \mathcal{N}(0, 9)$. 20 samples are uniformly drawn from a training interval $[-4, 4]$. 30,000 training examples were used for training a multi-layer perceptron (MLP) with a single hidden layer of 100 units. The model was tested with 10,000 samples drawn uniformly from $[-6, 6]$. Deterministic, VI, and partial Bayesian models with various $r_{bayes}$ were trained. Figure 2 illustrates how the partial Bayesian approach is capable of capturing the uncertainty with less than half of the parameters set as Bayesian.

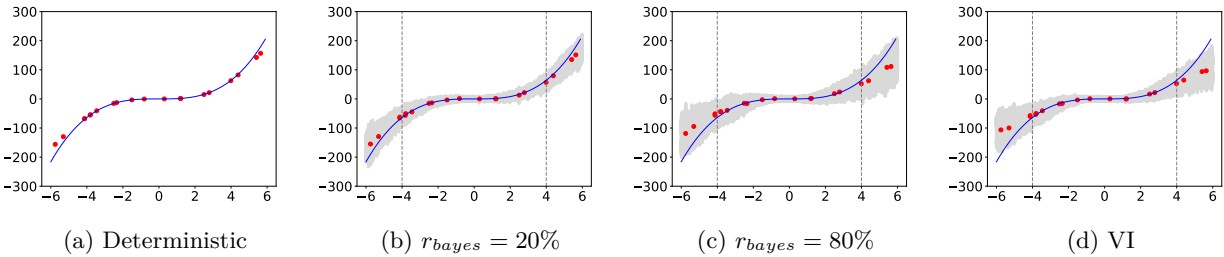

| (a) Deterministic | (b) $r_{bayes} = 20\%$ | (c) $r_{bayes} = 80\%$ | (d) VI |

Figure 2: Results on $y = x^3 + \epsilon$ toy dataset. Predictions on toy data, the blue line is ground truth $y = x^3$, red dots are sample test predictions, and the gray shaded area is the $3\sigma$ confidence interval. The vertical lines highlight the in- and out-of-distribution (ID, OOD) test points. The proposed model displays how varying $r_{bayes}$ allows the model to capture the predictive uncertainty for ID and OOD samples. A figure with a range of $r_{bayes}$ values is in Appendix: F.

## 4.2 MNIST Demo

For demonstration, we use a LeNet (LeCun et al., 1998) architecture to evaluate image classification performance on MNIST (LeCun, 1998). Figure 4a presents the test performance of various models as a function of the Bayesian parameter ratio ($r_{bayes}$). As $r_{bayes}$ increases while keeping the number of training steps fixed, we observe a decline in test performance, particularly in Brier Score.

Figure 3 visualizes the evolution of weight uncertainty (i.e., the $\sigma$ parameter) in the first layer of the network across different $r_{bayes}$ values. At $r_{bayes} = 5\%$, uncertainty is concentrated on central connections corresponding to key MNIST features. As $r_{bayes}$ approaches 100% (i.e., standard VI), uncertainty increases

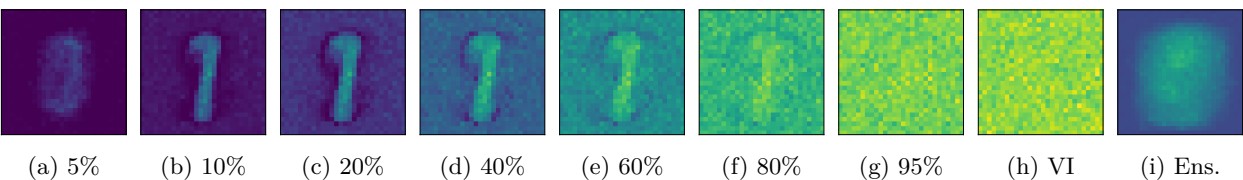

| (a) 5% | (b) 10% | (c) 20% | (d) 40% | (e) 60% | (f) 80% | (g) 95% | (h) VI | (i) Ens. |

Figure 3: MNIST Weight uncertainty ($\sigma$): heatmaps indicate the value of the weight uncertainty and highlight the location of the (selected) Bayesian weights in PBN as a function of $r_{bayes}$ (a-g), fully VI (h), and 5-member ensemble (i). The images highlight how partial Bayes focuses on the uncertainty of the important features within the given dataset.

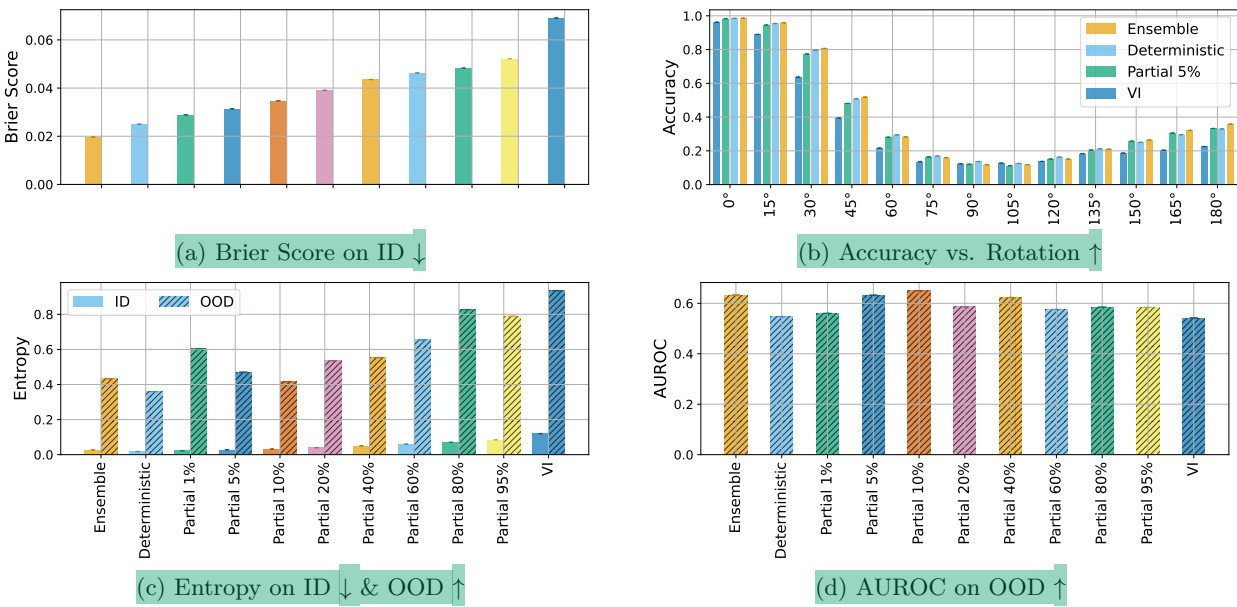

Figure 4: Results on MNIST: (a) Brier score ↓ for test set with increasing stochasticity, (b) Accuracy ↑ on Rotated MNIST (c) Entropy for MNIST (ID: in-distribution) ↓ fashionMNIST (OOD: out-of-distribution - hatched) ↑. (d) AUROC ↑ for OOD detection. (VI=Variational Inference)

approximately fourfold, introducing greater noise and instability—likely due to insufficient optimization steps for full convergence (see Appendix C for more details).

To assess out-of-distribution (OOD) generalization, we evaluate two scenarios: (1) covariate shift using rotated MNIST images in Figure 4b (2) dataset shift by training on MNIST and testing on FashionMNIST (far-OOD) in Figures 4c 4d. Under covariate shift, ensembles show greater robustness to perturbations, consistent with previous findings in Ovadia et al. (2019). When detecting far-OOD samples, AUROC scores indicate that Partial 10% performs on par with Ensembles, with both models outperforming other methods 4d.

### 4.3   Large Scale Networks

To assess whether our approach holds on larger-scale models such as ResNet18(He et al., 2016). As in previous sections, we demonstrate that very low percentages of Bayesian parameters exhibit competitive performance; therefore, in this section, we evaluate PBN with $r_{bayes}$ ≤5%. We train and evaluate ResNet18 on CIFAR10 with $r_{bayes} = 5\%$ on test data, and on corrupted data from CIFAR10-C (Hendrycks & Dietterich, 2019). Figure 5 demonstrates that PBN exhibits strong robustness to data corruptions, with performance degrading less than that of subnetwork inference via Linearized Laplace (Daxberger et al., 2021b) and other baseline methods. Notably, the robustness of PBN trained with variational inference (VI) aligns with the findings of Ovadia et al. (2019), which show that while VI models may converge to slightly lower ID accuracy, they tend to exhibit better stability under distributional shift.

Since the PBN approach relies on pre-trained point estimates to select the subnetwork, a natural question arises: does initializing the PBN with these deterministic weights also accelerate training? TTo investigate this, we compare the training behavior of a standard VI network with that of a PBN configured with $r_{bayes} = 5\%$. For the PBN model, we evaluate two initialization strategies: one where the network is initialized with deterministic weights, as described in section 3, and another where the trainable $\mu$ and $\rho$ parameters are randomly initialized.Training a ResNet-18 on CIFAR-10 across all three models reveals that PBN initialized from point estimates converges faster than both the randomly initialized 5% PBN (with the same number of variational parameters and distribution across the network) and the standard VI network.

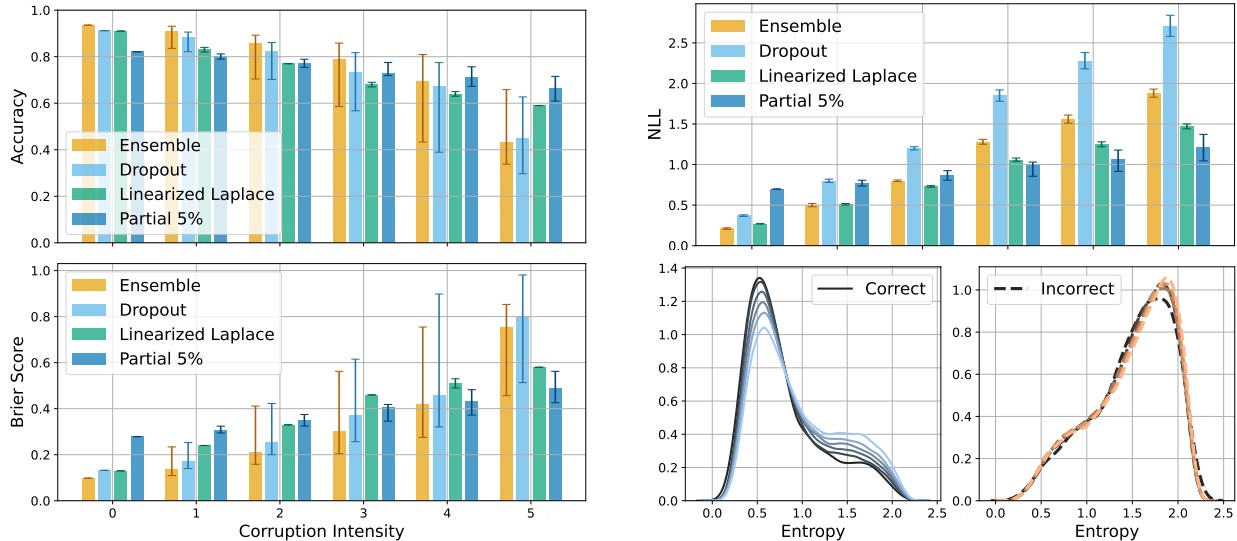

Figure 5: Calibration under covariate distributional shift of CIFAR10 with models: (ours) Partial with $r_{bayes} = 5\%$, Linearized Laplace (Daxberger et al., 2021b), ensemble, dropout. The x-axis is the corruption intensity of the CIFAR10 for 19 different corruptions in CIFAR10-C. (Bottom right) Entropy for correctly and incorrectly classified examples for both ID and OOD samples for proposed method (Partial VI) with $r_{bayes}$=5% . Baseline results retrieved from: Daxberger et al. (2021b) for Linearized Laplace, Ovadia et al. (2019) for Ensemble, Dropout, SVI.

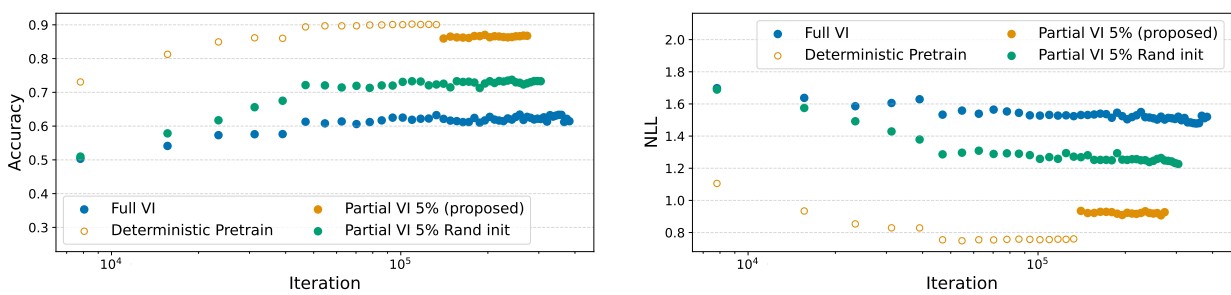

Figure 6: Test accuracy (a) and NLL (b) as a function of number of training iterations, for a full variational inference network (VI), a Partial Bayesian Neural Network with 5% Bayesian parameters initialized from deterministic weights (Partial VI 5% (proposed)), and another 5% partial Bayesian network initialized from random (Partial VI 5% Rand init). (ResNet18 on CIFAR10)

It also achieves lower NLL and higher accuracy, all while requiring significantly less training cost (see Figure 6).

We further evaluate the scalability of PBN on the ImageNet classification task (Deng et al., 2009) using ResNet50 (He et al., 2016). In contrast to standard VI (Ovadia et al., 2019) and linearized Laplace approximations (Daxberger et al., 2021b), which face challenges in scaling to large datasets, PBN approach with low Bayesian rates offers an effective trade-off between computational efficiency and reliable uncertainty estimation. As shown in Figure 7, PBN method maintains robustness under covariate shift on large-scale data, however it does not perform better than the baseline ensemble in this case (within error). However, Figure 7 demonstrates PBN exhibits increased predictive entropy with higher levels of corruption on ImageNet-C (Hendrycks & Dietterich, 2019), particularly for misclassified inputs, indicating well-calibrated uncertainty under distributional shift.

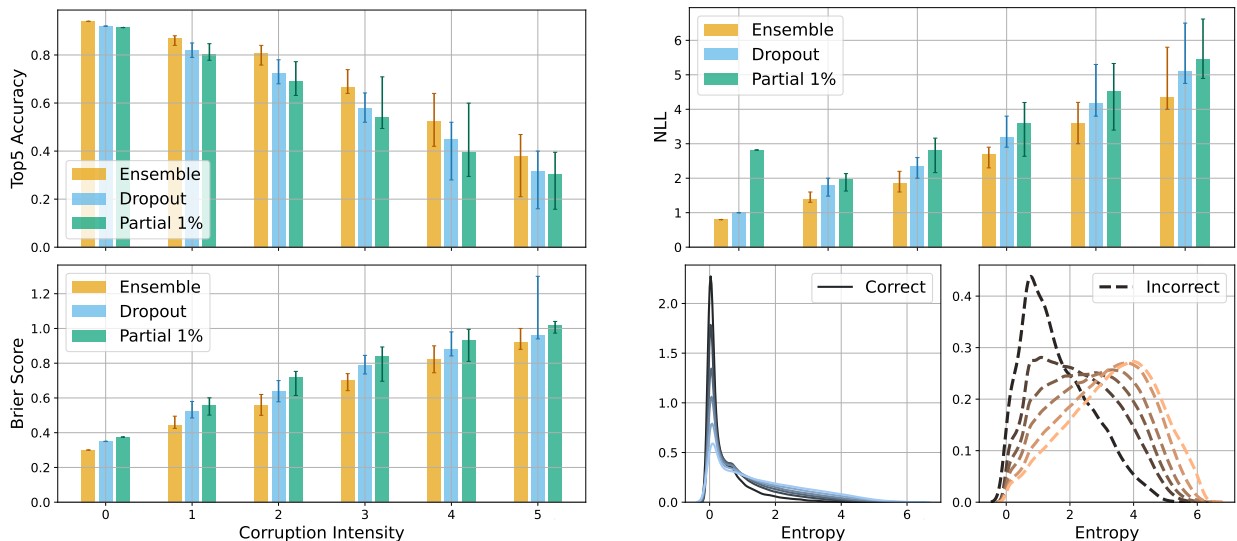

Figure 7: Results for ImageNet with respect to covariate shift performance. Top5 Accuracy, NLL, Brier Score, with Partial 1% displaying robustness to covariate shifts, with corruption intensity on x-axis. (Bottom right) Entropy values for correctly vs. incorrectly classified inputs for uncorrupted and corrupted data for Partial 1% model, displaying calibration under covariate shift, with increasing entropy (uncertainty) with increasing data corruption. Ensemble and Dropout results shown are retrieved from Ovadia et al. (2019)

## 4.4 Segmentation on CityScapes

We also demonstrate PBN's scalability on a multi-class pixel-level segmentation. Due to the high computational demands, general approaches for estimating uncertainty in segmentation are often limited to approximations such as ensembles and Monte Carlo dropout (Abdar et al., 2021) rather than variational Bayesian methods. Due to the seamless integration of PBN method, it enables uncertainty estimation for segmentation tasks. This contrasts with the method proposed by (Daxberger et al., 2021b), which utilizes linearized Laplace and selects the subnetwork based on the second-order Hessian. This process is computationally prohibitive for complex segmentation applications. We use a vanilla UNet architecture (Ronneberger et al., 2015) to demonstrate segmentation task on CityScapes Cordts et al. (2016), with 5 downward and

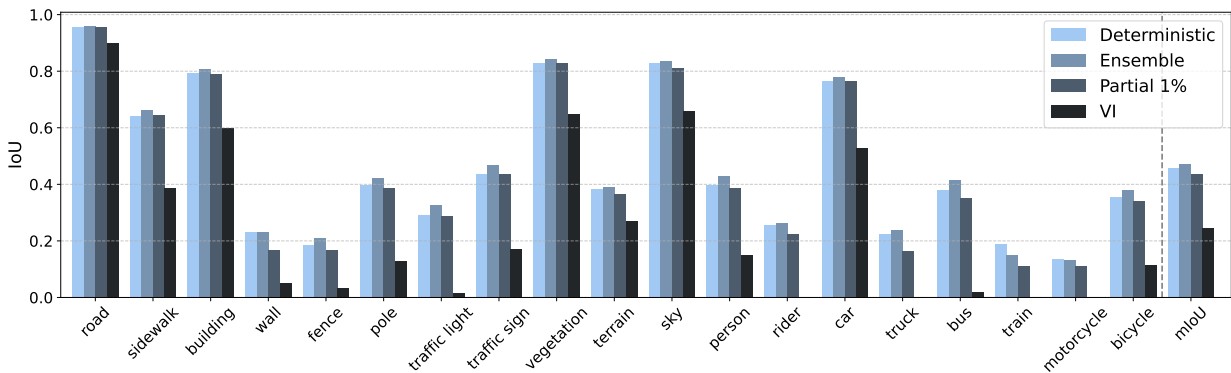

Figure 8: Comparative results on Cityscapes: per class IoU and mean IoU for each method. Partial method proposed with 1% variational parameters performing within 4% of Deterministic performance and 7% of Ensembles. On the other hand variational Bayesian method with 4-fold additional FLOPs performs 43% worse than Partial 1% model and 47% worse than Ensembles.

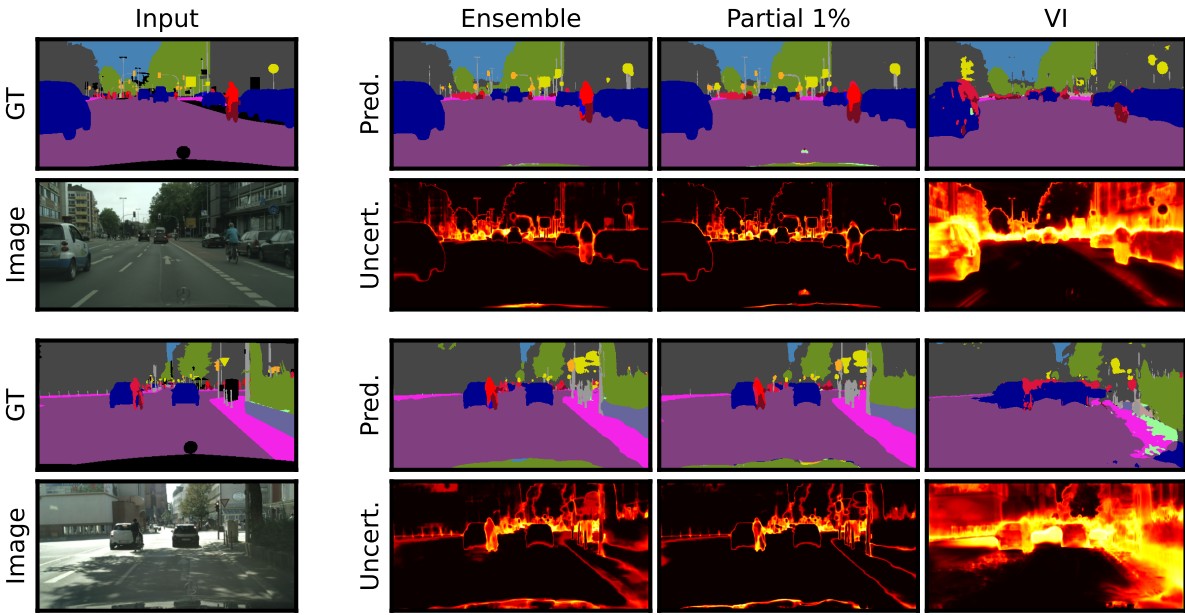

Figure 9: Cityscapes pixel-wise segmentation samples: from left to right: ground truth segmentation, segmentation and uncertainty for ensemble, Partial 1% (ours), and standard VI model. Low uncertainty is depicted as dark, while yellow-red indicates higher uncertainty. Note: the VI model is trained for an additional 4-fold number of iterations. Uncertainty is measured by entropy. (see additional qualitative examples in Appendix:10)

upward convolutional blocks, for a total number of parameters of over 31.38M. We compare the performance of the deterministic, VI, 5-member ensemble, and PBN with $r_{bayes} = 1\%$. Figure 9 shows the qualitative comparison of the ensembles, PBN-1%, and VI segmentation performance. Qualitatively, the PBN-1% segmentation uncertainty is consistent with errors in the prediction and at class boundaries. PBN-1% is also consistent with uncertainty estimations by ensembles, while utilizing $\sim 80\%$ fewer trainable parameters. Quantitatively, PBN has less than 5% degradation in IoU performance compared to the deterministic model, while the standard VI model performs very poorly with a 40% degradation in performance (Figure 8). The performance of the PBN on the segmentation task is both quantitatively and qualitatively comparable to that of ensembles. Examining the distribution of selected parameters in the subnetwork at low $r_{bayes} < 5\%$ aligns with (Sharma & Jennings, 2021), showing that the highest magnitude gradients are near the network edges, both input and output. Similarly, for ResNet18 with CIFAR-10, Bayesian weights are selected from layers near the input and output blocks, consistent with (Sharma & Jennings, 2021) findings. The distribution of the variational parameters as a function of layer depth is shown in Appendices 12 and 13.

## 5    Scope and Limitations

PBN method relies on pre-trained point estimates to initialize the Partial Bayesian Neural Networks, similar to (Daxberger et al., 2021b), however applied in the context of variational inference as opposed to Linearized Laplace. While the partial Bayesian formulation effectively reduces parameter complexity in large probabilistic models, its advantages diminish for smaller networks. As demonstrated in the toy example in Section 4.1, the added computational overhead may outweigh the benefits, particularly for uncertainty estimation in out-of-distribution (OOD) regions. For small to medium-sized models, ensembles remain more effective for both ID and OOD settings, as their relative computational cost is negligible. However, PBN displays advantages at larger networks and complex tasks, it serves as a practical solution, especially with its robustness to distributional shifts.

A current limitation of our approach is the reliance on posterior sampling via multiple forward passes to estimate uncertainty, as the method does not yet address the posterior sampling challenge inherent to Bayesian neural networks. In parallel, we are actively exploring improvements to reduce the sampling-related training and inference FLOPs in our implementation. Currently, the method supports linear and convolutional layers, making it suitable for regression, classification, and segmentation tasks. Extension to recurrent architectures remains an open direction for future work.

## 6 Conclusion

We demonstrated and examined a scalable algorithm for training partial Bayesian neural (PBN) networks using variational inference. Starting from a pre-trained deterministic model, PBN selects a subnetwork of variational Bayesian parameters, guided by first-order sensitivity analysis obtained through standard automatic differentiation—without additional computational overhead. This approach enables control over the degree of Bayesian modeling via a simple hyperparameter.

For small- to medium-scale networks ($10^4$–$10^5$ parameters), ensembles remain more effective in delivering well-calibrated predictive uncertainty. However, in larger models ($10^6$–$10^7$ parameters), PBN approach provides a cost-effective alternative for uncertainty quantification. With fewer than 5% of parameters designated as Bayesian, PBN approach achieves up to a 45% reduction in trainable parameters compared to a standard VI network and up to 80% compared to ensembles—while maintaining reliable uncertainty estimates.

Partial VI demonstrates strong robustness to data corruption and outperforms or matches established techniques such as ensembles, dropout, VI and linearized Laplace approximations (Daxberger et al., 2021b). Additionally, the method offers scalability to large-scale tasks, including classification on ImageNet and semantic segmentation, where traditional Bayesian neural network methods (e.g., VI or MCMC) often become infeasible. This work opens new avenues for efficient and scalable uncertainty estimation in deep learning, particularly in safety-critical domains where full Bayesian modeling remains impractical.

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

# A    Additional Segmentation Results

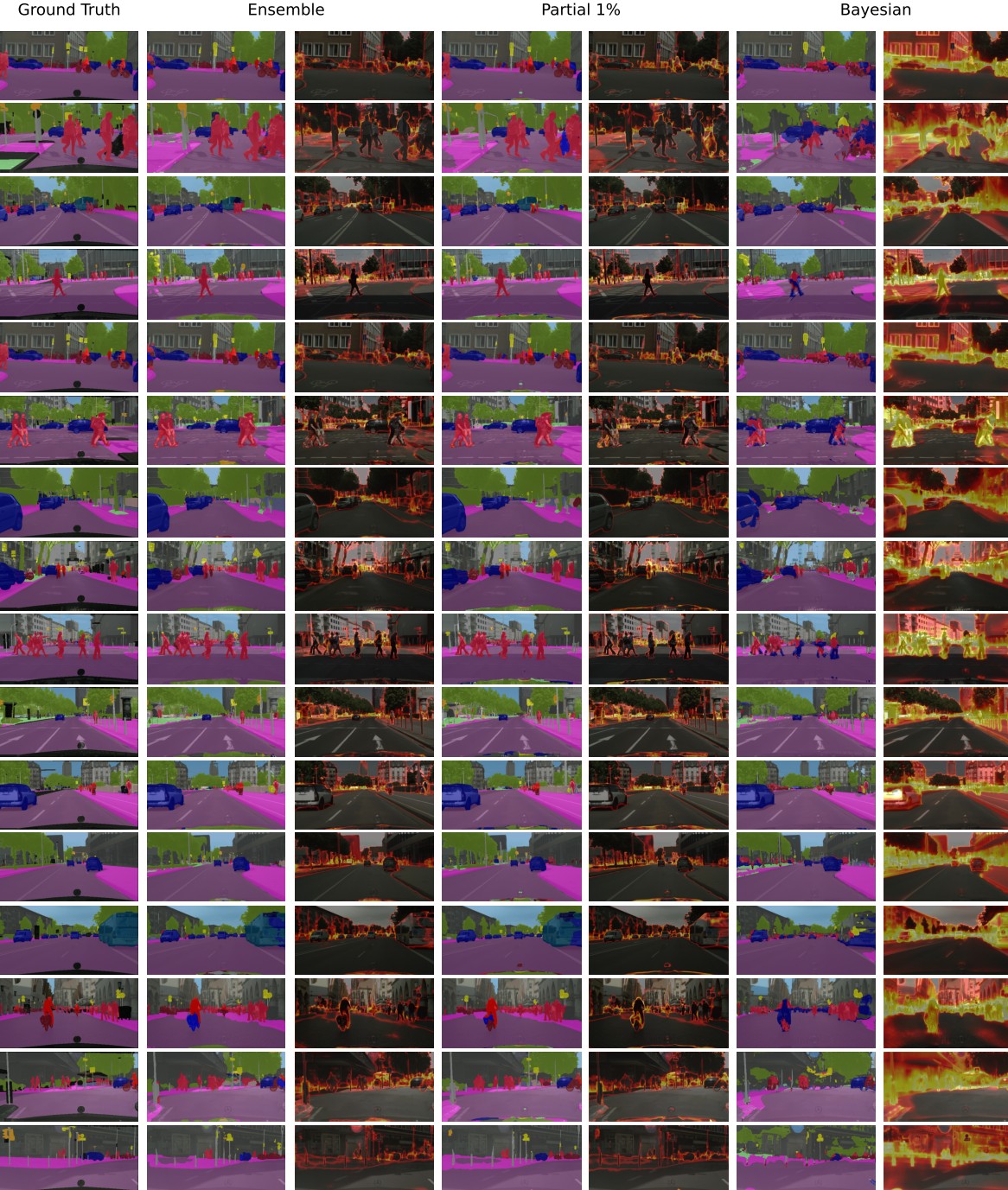

Figure 10: Cityscapes fine-segmentation samples: from left to right: ground truth segmentation, segmentation and uncertainty for ensemble, 1% PBN (ours), and standard VI model (Bayesian). Low uncertainty is depicted as dark overlays, while yellow-red indicates higher uncertainty. Note: the standard VI network is trained for an additional 4-fold Epochs.

# B  Parameter Complexity

Given a deterministic model of $\theta$ parameters as a baseline. A fully Bayesian model increases the number of parameters to $2\theta$, where each layer is characterized by two distinct parameters, accounting for the distribution's mean and standard deviation. For models employing partial Bayesian techniques, a singular Bayesian layer results in a parameter count of $\theta + L_{numel} < 2\theta$ where $L_{numel}$ represents the number of elements in a parameter layer. In partial Bayesian models with $n$ Bayesian weights, the parameter count becomes $\theta + n = (1 + r_{bayes})\theta$.

Table 1: Compute and memory costs relative to the deterministic model. $N$ denotes the number of posterior samples, $f_d$ denotes the FLOPs required for a single forward pass, $M$ is the number of ensemble members, and $m$ is the memory required to store a deterministic model ($m \sim f_d$) (Ovadia et al., 2019)

| Model | #Parameters | Relative FLOPS | Memory |
|---|---|---|---|
| Deterministic | $\theta$ | $3f_d$ | m |
| M-Ensemble | $M\theta$ | $3Mf_d$ | M×m |
| VI | $2\theta$ | $2 \times (N + 2)f_d$ | 2×m |
| PBN | $(1 + r_{bayes})\theta$ | $3f_d \times \text{epochs}_{pretrain} + (1 + r_{bayes})(N + 2)f_d$ | 2×m |

# C  FLOPs Count

The floating point operations (FLOPs) are computed as the sum of tensor additions and multiplications. The FLOPs for a forward pass include calculating the loss for a single batch for the given set of parameters $\theta_i$ at epoch $i$. In the backward pass, the loss is used to calculate the gradients of the parameters $\nabla\theta_i$ and the gradient of the activations. Therefore, to account for the total FLOPs for a single forward-backward pass, it would cost $3f_d$ for a single sample, where $f_d$ is the number of FLOPs for a fully-dense deterministic forward pass for a given architecture. For a given architecture, we can compute the following:

- **Ensemble** Given $M-$ensemble members, the cost for a single sample scales with $3 \times M \times f_d$.

- **VI** Given $N-$posterior samples, the cost for a single sample scales with $2 \times N \times f_d + 2(2 \times f_d)) = 2 \times f_d(N + 2)$, where the $2\times$ term accounts for the addition of the uncertainty $\sigma$ parameter.

- **Partial Bayesian** Given a Bayesian rate $0 < r_{bayes} < 1$ the number of FLOPs for the forward method is $(1 + r_{bayes}) \times N \times f_d$, where $N$ is the number of posterior samples, and backward method costs $2 \times (1 + r_{bayes}) \times f_d$ for a total of $(1 + r_{bayes})f_d(N + 2) + 3f_d \times \text{epochs}_{pretrain}$ where the second term accounts for the deterministic pre-training step. FLOPs associated with the intermediate sensitivity analysis step (algorithm 1 line 8) are not taken into account as the contribution to the total FLOPs is negligible.

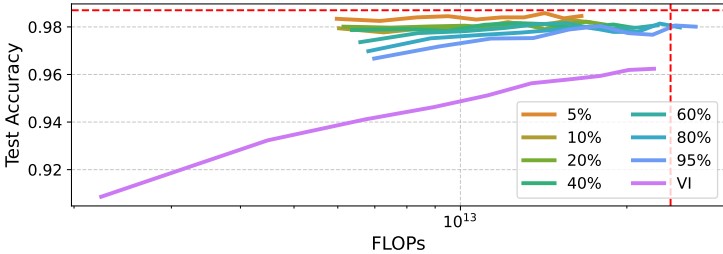

Figure 11: Test Accuracy vs. training FLOPs with the ensemble performance highlighted with the intersection of the dotted red lines for MNIST. The Legend = $r_{bayes}$ values, and Bayes for the fully Bayesian model.

# D Sparse Bayesian Parameter Distribution - ResNet

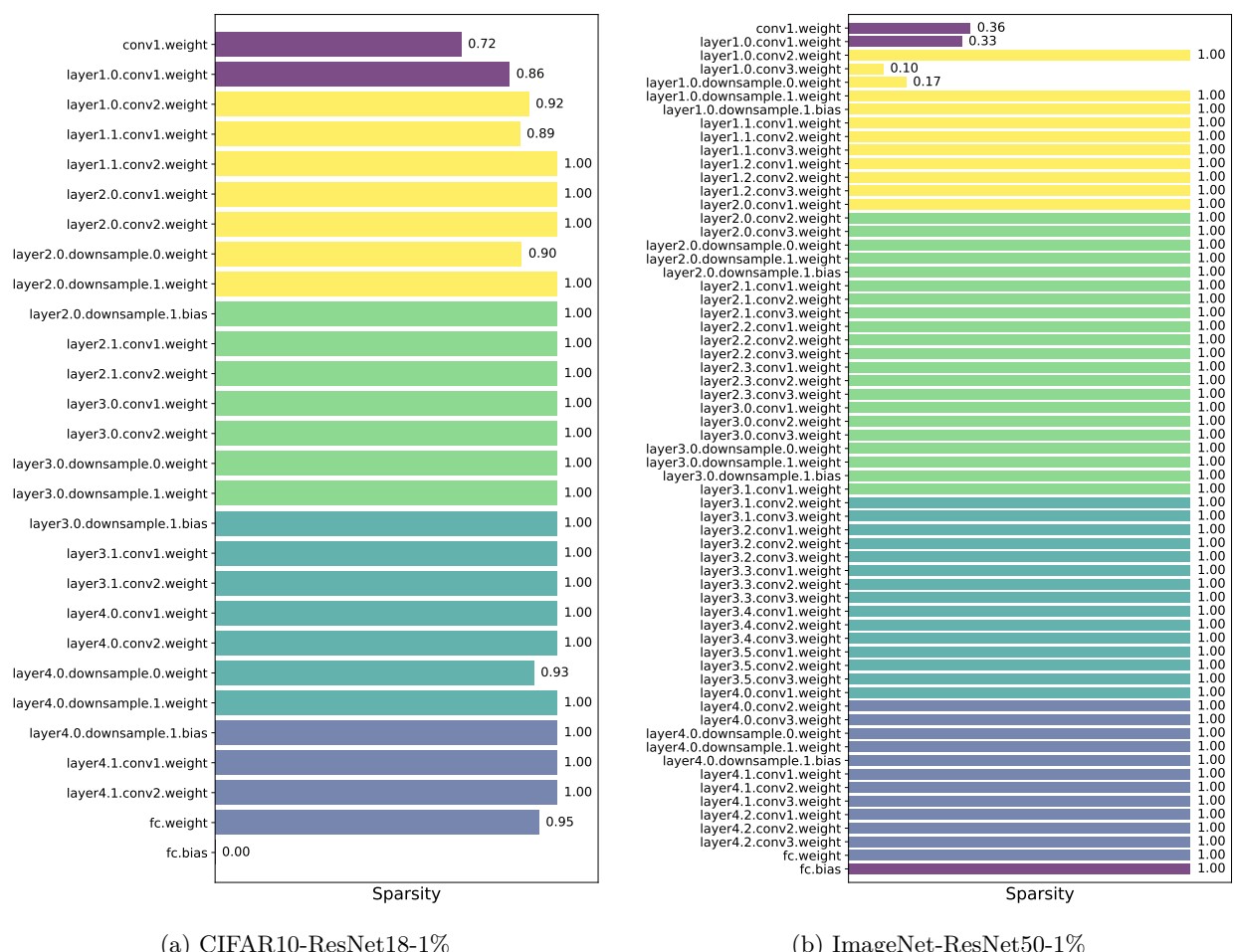

(a) CIFAR10-ResNet18-1%  (b) ImageNet-ResNet50-1%

Figure 12: Sparsity values of ResNet18(50) layers with $r_{bayes} = 1\%$]. Layers with a sparsity value of 0.99-1.0 are fully deterministic.

# E    Sparse Bayesian Parameter Distribution - UNet

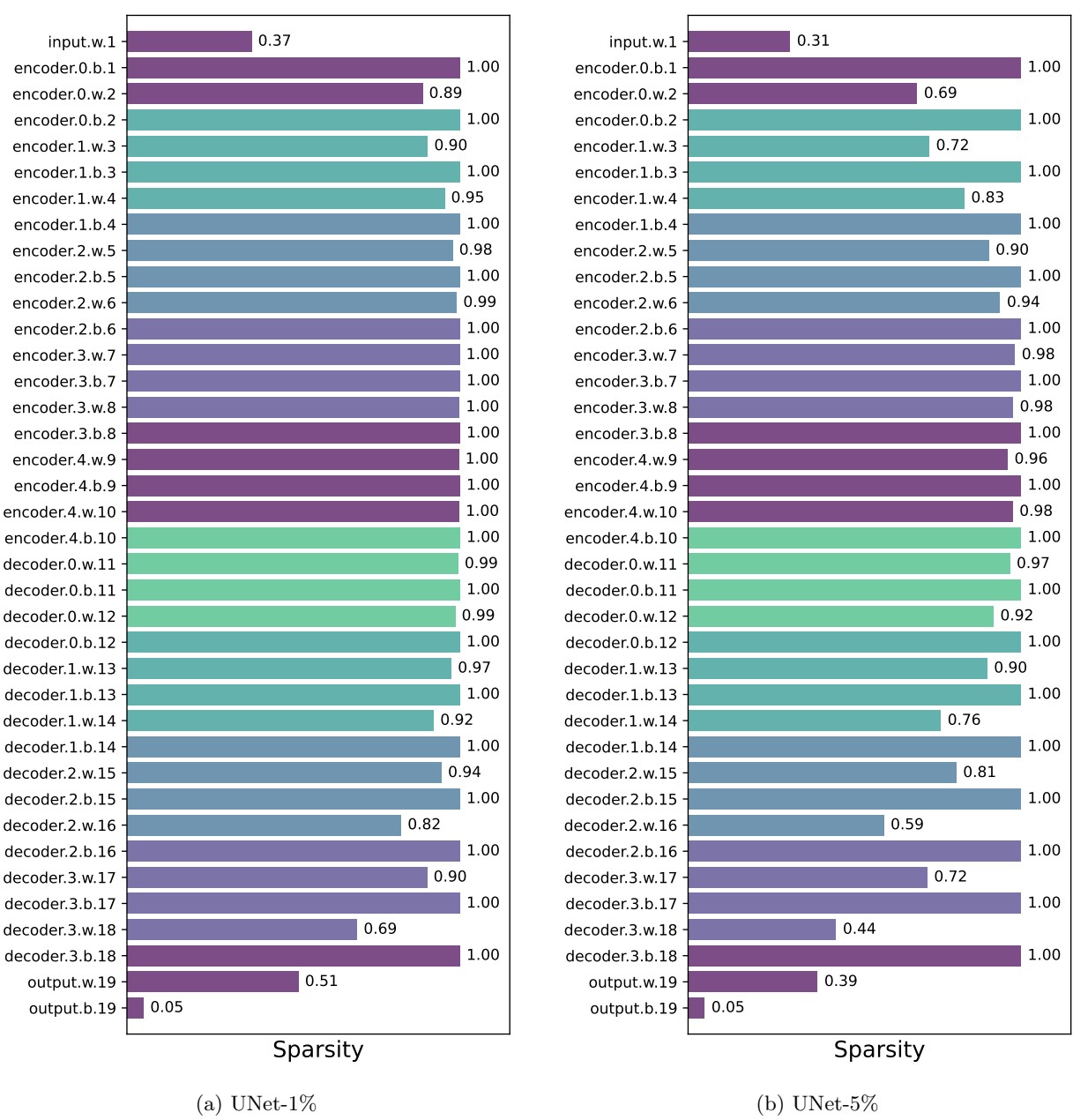

(a) UNet-1%                    (b) UNet-5%

Figure 13: Sparsity values of UNet with encoder/decoder blocks of feature [64, 128, 256, 512, 1024] layers with $r_{bayes} = [1\%, 5\%]$. Layers with a sparsity value of 0.99-1.0 are fully deterministic.

## F Toy Regression Dataset

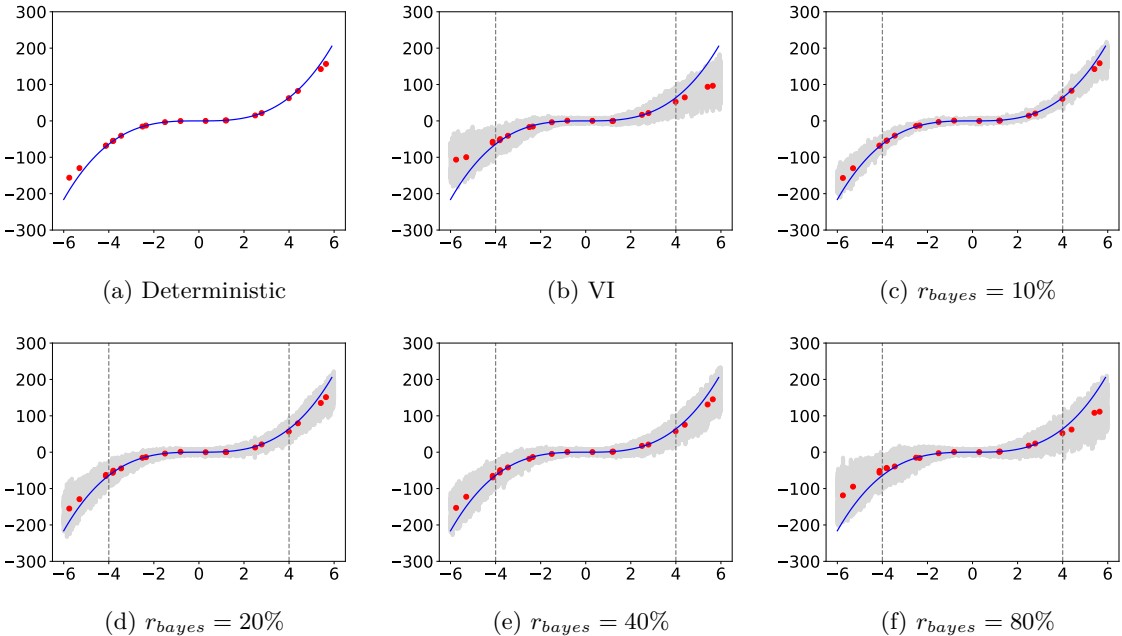

(a) Deterministic      (b) VI      (c) $r_{bayes} = 10\%$

(d) $r_{bayes} = 20\%$      (e) $r_{bayes} = 40\%$      (f) $r_{bayes} = 80\%$

Figure 14: Results on $y = x^3 + \epsilon$ toy dataset. (a-d) Predictions on toy data, the blue line is ground truth $y = x^3$, red dots are sample test predictions, and the gray shaded area is the $3\sigma$ confidence interval. Predictions on (a) Deterministic, (b) Standard VI Bayesian approach, (c-f) Partial Bayesian models with different Bayes rates. Note that the dotted lines highlight the in- and out-of-distribution test points. The partial Bayes model displays how varying $r_{bayes}$ allows the model to capture the predictive uncertainty for in- and -out of distribution samples.

## G   Learned Weight Distribution

We evaluated the weight distribution using a Gaussian Mixture Model (GMM) for the 5-member ensemble and PBN-1%. Figure 15a shows that the ensemble's weight distribution is slightly narrower than that of the PBN-1%. A Kolmogorov–Smirnov (KS) test was used to quantify the difference between the two GMM distributions. Figure 15b plots the cumulative distribution functions (CDFs) of the weight GMMs. The KS statistic for PBN-1% compared to the 5-ensemble is 0.07 (with p-value $< 1e - 10$), indicating a high similarity between the two distributions.

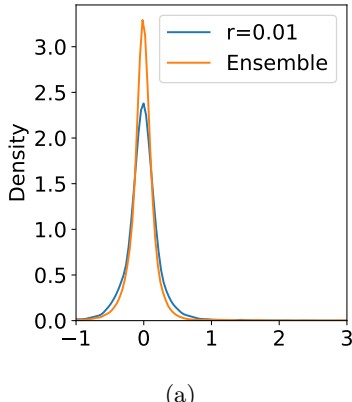
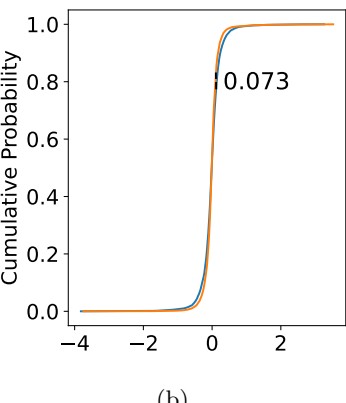

(a)                                             (b)

Figure 15: (a) Gaussian Mixture Model (GMM) for CIFAR10 ResNet models from PBN-1%, 5-member ensemble. (b) Kolmogorov–Smirnov (KS) cumulative distribution function for PBN-1% with a KS statistic of 0.07 and a p-value $< 1e - 5$. The KS-test provides supporting evidence for the similarity of the learned models by assessing the similarity of their weight distributions.

## H   Additional Model and Training Details

### H.1   Hardware and Software

Models were trained with an NVIDIA RTX A6000 GPU with Pytorch 2.0.1 version.

### H.2   Model Details

#### H.2.1   MNIST

Standard LeNet 300-100 network was used for the MNIST classification experiments with a batch size of 50. SGD optimizer with a fixed learning rate of 0.01, a momentum of 0.9, and an L2 regularization coefficient of $10^{-5}$. ReLU non-linearities were used for all models and batch normalization. Models trained for 50 epochs.

Table 2: List of datasets used in the paper, with the train/test splits.

| Dataset | Source | Size (Train, Test) | Input Size | #Classes |
|---|---|---|---|---|
| **MNIST** | (LeCun, 1998) | 60K , 10K | 28×28 | 10 |
| **Fashion-MNIST** | (Xiao et al., 2017) | 60K, 10K | 28×28 | 10 |
| **CIFAR10** | (Krizhevsky et al., 2009) | 50K, 10K | 32×32 | 10 |
| **CIFAR10-C** | (Hendrycks & Dietterich, 2019) | 10K | 32×32 | 10 |
| **SVHN** | (Netzer et al., 2011) | 732K, 260K | 32×32 | 10 |
| **CityScapes** | (Cordts et al., 2016) | 4500, 500 | 256×512 | 20 |

### H.3 CIFAR-10/SVHN

#### H.3.1 Small ResNet

A small ResNet with 4 ResBlock was used for initial experiments, containing features of [16, 32, 64]. Each ResBlock contains 2 convolutional layers, one with kernel size 3, the first with a stride of 2 and the second convolutional layer with a stride of 1. The input convolution is of kernel size 3 and a stride of 2, followed by a Maxpool layer of (2,2). Similarly, as above, a batch size of 50, an SGD optimizer with a fixed learning rate of 0.01, momentum of 0.9, and L2 regularization coefficient of $10^{-5}$. Models were trained for 50 epochs. Both CIFAR10 and SVHN datasets were trained with $r_{bayes} = [0.1\%, 0.5\%, 1\%, 5\%, 10\%, 20\%, 40\%, 60\%, 80\%, 95\%$.

#### H.3.2 Standard ResNet18-V1

Our second CIFAR10 model used a standard ResNet18-V1 (He et al., 2016) with ReLU activations. The max pool layer was removed, and the initial convolution of 7x7 of stride 2 was changed to 7x7 with a stride of 1 to avoid significant downsampling of the smaller CIFAR10 images. The model was trained with a batch size of 200 for 50 epochs using an SGD optimizer with a momentum of 0.9 and a learning rate schedule that includes a linear warm-up for up to 5 epochs to a maximum learning rate of 1.6. Then, the learning rate is reduced by 10 at epoch 30. L2 regularization coefficient of $10^{-4}$ was used. For the Bayesian and Partial Bayesian approaches, the models were trained for 100 epochs. A dropout rate of 0.2 was used to avoid overfitting; dropout layers were added after the FC layer and after the first convolutional layer in each convblock.

### H.4 CityScapes

We use a standard UNet (Ronneberger et al., 2015) with standard encoder-decoder architecture, convolutional blocks with feature output sizes of [64, 128, 256, 512, 1024]. The upward paths between each block in the decoders are bilinear interpolations due to a limitation in the Pytorch backward pass customization for TransposeConvolutional layers. The deterministic models were trained with SGD optimizer, a momentum of 0.9, and an initial learning rate of 0.1, which is reduced by 0.5 on plateauing validation performance. The models were trained for 150 epochs. The data was resized to 256×512. A weighted cross-entropy loss was used to train the dataset to balance the minority classes, such as persons, riders, trailers, motorcycles, caravans, trains, etc. We follow the recommended data labeling setup by the dataset initial release (Cordts et al., 2016).

### H.5 Ensembles

Five member ensembles were used in our experiments, where each member was trained independently with a different random seed [0,1,2,3,4] to ensure reproducibility.

### H.6 Variationa Inference

For all Partial Bayesian training or standard VI training -ELBO loss is minimized, where $\mathcal{L} = \mathcal{L}(f_{\Theta_d}(x_i), y_i) + \beta \cdot KL(\mathcal{N}_{prior}, \mathcal{N}_{post})$, the first term is the MLE term (for regression it's MSE for classification it's cross-entropy loss), $\beta$ is the weight of the KL divergence term. The KL weight term, $\beta$, is annealed during training epochs, transitioning smoothly from an initial value [0.2] to a desired target value [0.01] as a function of epochs $\beta_i = \beta_{target} - (\beta_{target} - \beta_{init}) \times (epoch_i/epoch_{total})$. We experimented with fixed vs. different annealing schedules and found this schedule optimal to minimize the KL divergence in the first few epochs and then optimize the task-specific loss. 5 posterior samples were drawn during training and inference.

## I Visualization of Sparse Gradient Updates

To train layers with a mix of deterministic and variational parameters the layers are redefined. Consider a 3×3 weight matrix: $\mu$ parameter is initialized from the deterministic point estimate values, $\sigma$ parameter is initialized for the parameters associated with $\text{Topk}(|\nabla_\theta|)$ - see 16.

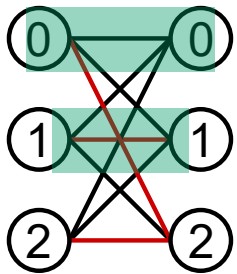 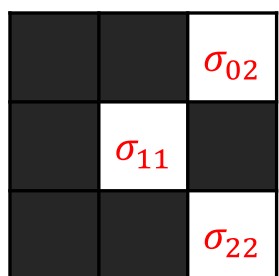

| $\mu_{00}$ | $\mu_{01}$ | $\mu_{02}$ |
|---|---|---|
| $\mu_{10}$ | $\mu_{11}$ | $\mu_{12}$ |
| $\mu_{20}$ | $\mu_{21}$ | $\mu_{22}$ |

| | | $\sigma_{02}$ |
|---|---|---|
| | $\sigma_{11}$ | |
| | | $\sigma_{22}$ |

| $\delta_{00}$ | $\delta_{01}$ | $\mathcal{N}_{02}$ |
|---|---|---|
| $\delta_{10}$ | $\mathcal{N}_{11}$ | $\delta_{12}$ |
| $\delta_{20}$ | $\delta_{21}$ | $\mathcal{N}_{22}$ |

Figure 16: Visual representation of a 2D array for a layer in a partial Bayesian model, with $\mu$ (left), sparse $\sigma$ (middle), resulting parameter type (point vs. distribution) (right). A dark element within the $\sigma$ parameter indicates where a connection is represented with a point (deterministic) rather than a distribution (Bayesian). The density of the Bayesian connections is controlled by the hyperparameter $r_{bayes}$.

## J Derivation of Variance-Sensitivity Relationship in Section 3

This section includes the full derivation of the relationship between this important measure of uncertainty, sensitivity, and the uncertainty in the output as briefly shown in 3. For the network $f$, parameterized by $\Theta$ (for brevity and ease of readability, we will refer to $f_\Theta(x)$ with $f(\Theta)$ as $\Theta$ is the parameter of interest in this case), the variance in the output is defined as follows,

$$
\begin{aligned}
Var(y) &= \mathbb{E}\left[(f(\Theta) - \mathbb{E}[f(\Theta)])^2\right] \\
&= \mathbb{E}\left[f(\Theta)^2 - 2f(\Theta)\mathbb{E}[f(\Theta] + (\mathbb{E}[f(\Theta)])^2\right] \\
&= \mathbb{E}\left[f(\Theta)^2\right] - 2\mathbb{E}[f(\Theta)]\mathbb{E}[f(\Theta)] + (\mathbb{E}[f(\Theta)])^2
\end{aligned}
\tag{7}
$$

$$
Var(y) = \mathbb{E}\left[f(\Theta)^2\right] - (\mathbb{E}[f(\Theta)])^2
\tag{8}
$$

We can derive approximations for the terms $\mathbb{E}[f(\Theta)^2]$ and $(\mathbb{E}[f(\Theta)])^2$ by performing a Taylor series expansion of $f(\Theta)$ about the mean parameter value $\mu_\Theta$. The second term is derived by the Taylor series expansion of the first moment about $\mu_\Theta$:

$$
\begin{aligned}
\mathbb{E}[f(\Theta)] &= \mathbb{E}[f(\mu_\Theta + \Theta - \mu_\Theta)] \\
&\approx \mathbb{E}\left[f(\mu_\Theta) + f'(\mu_\Theta)(\Theta - \mu_\Theta) + \frac{1}{2}f''(\mu_\Theta)(\Theta - \mu_\Theta)^2\right] \\
&= f(\mu_\Theta) + f'(\mu_\Theta)\mathbb{E}[\Theta - \mu_\Theta] + \frac{1}{2}f''(\mu_\Theta)\mathbb{E}\left[(\Theta - \mu_\Theta)^2\right] \\
&\text{given } \mathbb{E}[\Theta - \mu_\Theta] = 0 \text{ and } \mathbb{E}\left[(\Theta - \mu_\Theta)^2\right] = \sigma_\Theta^2
\end{aligned}
\tag{9}
$$

$$
\therefore \mathbb{E}[f(\Theta)] = f(\mu_\Theta) + \frac{1}{2}f''(\mu_\Theta)\sigma_\Theta^2
\tag{10}
$$

For the second term, $\mathbb{E}[f(\Theta)^2]$ let $g(\Theta) = f(\Theta)^2$ to simplify the Taylor series expansion to the same form as above, for a first moment approximation of $\mathbb{E}[g(\Theta)]$ as in equations 10

$$\mathbb{E}[g(\Theta)] = \mathbb{E}\left[g(\mu_\Theta + \Theta - \mu_\Theta)\right]$$

$$\approx \mathbb{E}\left[g(\mu_\Theta) + g'(\mu_\Theta)(\Theta - \mu_\Theta) + \frac{1}{2}g''(\mu_\Theta)(\Theta - \mu_\Theta)^2\right]$$

(11)

such that

$$g'(\mu_\Theta) = 2f(\mu_\Theta)f'(\mu_\Theta) \text{ and } g''(\mu_\Theta) = 2\left(f'(\mu_\Theta)^2 + f(\mu_\Theta)f''(\mu_\Theta)\right)$$

$$\therefore \mathbb{E}[f(\Theta)^2] \approx \mathbb{E}\left[f(\mu_\Theta)^2 + 2f(\mu_\Theta)f'(\mu_\Theta)(\Theta - \mu_\Theta) + \left(f'(\mu_\Theta)^2 + f(\mu_\Theta)f''(\mu_\Theta)\right)(\Theta - \mu_\Theta)^2\right]$$

$$\therefore \mathbb{E}[f(\Theta)^2] = f(\mu_\Theta)^2 + \left(f'(\mu_\Theta)^2 + f(\mu_\Theta)f''(\mu_\Theta)\right)\sigma_\Theta^2$$

(12)

Then by substituting equations 10, 12 into equation 8:

$$\therefore Var(y) \approx f(\mu_\Theta)^2 + \left(f'(\mu_\Theta)^2 + f(\mu_\Theta)f''(\mu_\Theta)\right)\sigma_\Theta^2 - f(\mu_\Theta)^2 - \frac{1}{4}f''(\mu_\Theta)^2\sigma_\Theta^4 - f(\mu_\Theta)f''(\mu_\Theta)\sigma_\Theta^2$$

$$= f'(\mu_\Theta)^2\sigma_\Theta^2 - \frac{1}{4}f''(\mu_\Theta)^2\sigma_\Theta^4$$

$$\approx f'(\mu_\Theta)^2\sigma_\Theta^2$$

(13)

$$\therefore \sigma_y^2 \equiv \left(\frac{\partial f}{\partial \Theta}\right)^2\sigma_\Theta^2$$

