# OpenReview forum: "Efficient Uncertainty Estimation via Sensitivity-Guided Subnetwork Selection for Scalable Variational Inference"
_TMLR — Rejected by TMLR_

### Review · Reviewer_QJx7 · 2025-06-05

**Summary Of Contributions:**

This paper introduces a method of subnetwork selection, wherein they choose parameters that most directly influence the network output, i.e. parameters with the greatest network gradient magnitude. They then apply mean field variational inference to this subnetwork, in order to approximate the network posterior. The performance of this subnetwork MFVI is evaluated against full VI as well as other Bayesian methods.

**Audience:**

No

**Claims And Evidence:**

No

**Requested Changes:**

Currently, this paper is not worthy of submission. The contribution of this work seems very minor in comparison to previous works. The quality of the submission is far from a publishable level.

If you are seeking to prepare your paper for a future submission, addressing the above issues would be key. Some examples of smaller issues would be:

- Please place brackets around references, as was asked in the TMLR style sheet, unless the reference is in-line (though the majority of your references were not in-line, and require bracketing).
 - Discussion of leading Bayesian methods needs to be improved.
 - Related works subsection “computational demands of BNNs’ seems to miss key speedups, such as SWAG, Last-layer LLA, BatchEnsemble, Sampling-LLA etc.
 - Use of ‘second-order’ or ‘higher-order’ gradients seems incorrect.
 - The sensitivity-analysis section was under-whelming. This theory needs to be tightened to the core relationship, which is simply that you are taking the parameters which enact the most change in the output, and you are using this sensitivity analysis to show that the gradient values decide this.  Interestingly, this seems to be the inverse method to [3], where they take parameters with maximum marginal variance, as measured by the inverse of the GGN-Hessian, i.e. inverse of the gradient squared.
 - k=Topk(|∇_θ |,k) is a meaningless notation.
 - Method section needs a lot of work. For example, the number of datapoints and the number of parameters is both given by N. The notation section shouldn’t describe the MFVI procedure. Equation (1) is meaningless without saying what x,y are (only x_i, y_i are described previously).
 - Introduce the standard Bayesian notation of the likelihood and prior, and explain MFVI. Explain how MFVI is trained, and introduce the standard parameters.
 - Notation of ρ,μ_posti,σ_posti,μ_{ij}  was very confusing. What is i,j here? Also if you set ρ=0 for the deterministic parameters, won’t σ=ln⁡2?
 - Is Entropy of Expectation summed over all classes c?
 - Claim of 50% and 80% reduction don’t make sense. Can’t you choose the number of parameters? At 1%, wouldn’t the reduction be 98%, as there are mean and variance for every ‘Bayesian’ parameter? Wouldn’t the ensemble reduction be simply 5x the reduction for a single network? You also claim at one point that reduction is > 95%.

**Strengths And Weaknesses:**

There is a major issue with this paper: the similarity of this paper to [2] needs to be discussed. It seems to me that the current submission borrows the majority of its work from [2], and provides no new frameworks or methods. The contribution seems to only be an empirical investigation of PBN for common UQ datasets and problems, versus the medical datasets used in [2] (i.e. MNIST vs Chest-MNIST). If this is the case, then this seems a very minor contribution, and to me does not warrant publication.

## Strengths

The main method of subnetwork selection is quite interesting. It intuitively makes sense, as it is selecting the most important parameters to the network. It seems that this simple idea is novel in the UQ community. As subnetwork selection is promising, new techniques to select this subnetwork are important to analyse.

The performance of this method in comparison to MFVI (which is known to perform poorly) is very interesting as well; this method seems to prevent overfitting of the new approximate posterior q(theta). The results in Figure 8 and Figure 9 are promising and imply that only modelling the variance of a small subset of the parameters may enable uncertainty quantification performance similar to Deep Ensembles.

## Weaknesses

Unfortunately, this paper has many critical issues.

The focus of this paper is lacking. The main objective seems to be to “investigate the convergence behaviour of…PBN… under varying levels of model stochasticity”, and while this objective is achieved, a lot of space goes into comparing this PBN to other Bayesian methods. This would be a fine objective in and of itself, however the comparison to other methods is lacking in depth, and so the experiments seem to half-accomplish two goals.

Comparison to other methods is lacking due to poor experimental details, and poor experimental choice. Figure 2 should employ the much more recent toy regression test of [1], where in-between and far-from data uncertainty is tested. As it stands, it is hard to determine which model has the ‘best’ uncertainty; shouldn’t the variance be nearly uniform, as all training points are spread nearly uniformly? In Figure 3 it is unclear where on the picture that the uncertainty should be greatest. I do not believe the interior of the number should be uniformly uncertain; perhaps evaluating a number that could be both, say a ‘1’ and a ‘7’, and seeing if the uncertainty is greatest at the top of the digit. The results given in Figure 4 seem dubious. Firstly, the y-axis values in (a) and (b) seem miniscule for MNIST. A mean test Brier score of 1e-12 is very tiny. Secondly, the Brier score in (d) for 0 degrees rotation has ensemble smaller than partial 5%, yet in (a), which should be the same as (d) for 0 degrees rotation, has mean ensemble score greater than partial 5%.  For Figure 5, I believe that plotting relative error is more apt for the appendix. While it is interesting that the relative error is smaller for Partial 5% than the other methods, I more care about the absolute error. It seems that a metric which shows MC-Dropout outperforming the gold standard Deep Ensembles is not a very enlightening metric. Also, the authors will need to provide the details of the subnetwork LLA method. The same issues arise for Figure 6.

The writing of the paper needs improvement. Currently, it is not at a level which can be published. Many sentences are unclear or incorrectly written, there are repeating ideas that could be succinctly explained once, figures are sometimes introduced out of order, ideas in a paragraph are not always connected, there are uncommon uses of phrases, such as “second-order gradient”, notation is non-uniform, undefined or confusing, references have no bracketing, etc.

Computational speedup of this method versus other post-hoc methods is over-estimated. The subspace-LLA method [3] can employ SWAG-diagonal to find a subnetwork, yet the paper only describes the method of storing and computing the full inverse GGN-Hessian. SWAG-Diagonal does not have a very large cost, and so subspace-LLA should not be too far from this method in computational demand. Even using LLA on the last layer of a network is relatively well-performing, with minimal computation cost.

Further, the paper writes off post-hoc methods for subnetwork inference by stating of them: “this approach is limited as it relies on training a dense BNN as a first step and is followed by a post-hoc analysis for pruning “. Is this not exactly what the PBN method does?

Comparison to other subspace uncertainty quantification methods needs to be looked at, such as the PCA method of [4] and [3], and how your subspace selection technique improves/detracts performance compared to theirs.

References:

[1] Foong, Andrew YK, et al. "'In-Between'Uncertainty in Bayesian Neural Networks."

[2] Abboud, Zeinab, Herve Lombaert, and Samuel Kadoury. "Sparse bayesian networks: Efficient uncertainty quantification in medical image analysis."

[3]  Daxberger, Erik, et al. "Bayesian deep learning via subnetwork inference."

[4] Izmailov, Pavel, et al. "Subspace inference for Bayesian deep learning."

---

> ### Author Response · Authors · 2025-07-10
> **Response1**
>
> We thank the reviewer for their comments and feedback for improving the manuscript. Please find our responses to the questions/comments:
>
> **Comments**
>
> > 1. The focus of this paper is lacking.
>
> We agree that clarity around the paper’s objective is important and appreciate the opportunity to clarify our intent. The goal of the paper is to evaluate the scalability and practicality of partial Bayesian network (PBN) through subnetwork variational inference for uncertainty estimation in large-scale and diverse tasks. While we do investigate the convergence behavior of PBNs under varying levels of model stochasticity,  this is not the sole focus. Rather, it is one part of a broader study aimed at understanding whether PBNs can serve as an effective and computationally efficient alternative for uncertainty modeling.
>
>    The paper includes additional essential experiments, particularly robustness under distributional shifts and OOD data, because these are critical for assessing the reliability of the uncertainty estimates. Without such tests, any uncertainty measure would be incomplete and potentially misleading in safety-critical scenarios. Comparisons to existing Bayesian approaches (e.g., Dropout, Ensembles, VI, subnetwork via Linearized Laplace) are included not as a standalone benchmarking goal, but to position PBNs within the landscape of uncertainty estimation methods and highlight their strengths in terms of scalability and calibration. While we recognize that the experiments span multiple axes (e.g., convergence, robustness, scalability), they are unified by a single purpose: to evaluate the practical effectiveness of PBNs for scalable uncertainty estimation. We have clarified the goals/questions to be answered in Section 4.
>
> > 2. Comparison to other methods is lacking
>
> We have evaluated our method across multiple tasks, with a particular emphasis on scalability, as evidenced by results on large-scale datasets such as ImageNet and Cityscapes. Our comparisons include state-of-the-art dense uncertainty estimation methods, such as ensembles, standard variational inference (VI), and Dropout, as well as a more closely related two-stage approach: subnetwork training followed by inference using the Linearized Laplace (LLA) [1]. Our results demonstrate that the proposed method scales effectively in scenarios where approaches like LLA and standard VI struggle, and where ensembles become prohibitively expensive to train. On the CIFAR-10 benchmark, our method achieves superior robustness to data corruptions (Figure 5) while maintaining significantly lower parameter complexity and employing a simpler, more practical subnetwork selection strategy.
>
> > Figure 3 it is unclear where on the picture that the uncertainty should be greatest.
>
> On Figure 3 for Weight uncertainty visualization of MNIST: To clarify, as stated in the caption, Figure 3 visualizes \textit{ weight uncertainty}, not predictive uncertainty. These images are intended to show where the Bayesian parameters are concentrated in the input space. For lower Bayesian parameter rates, the uncertainty is centered around the digit, indicating that the method identifies and focuses on regions corresponding to salient input features. While this visualization is not meant to capture instance-specific ambiguity (e.g., distinguishing between a '1' and a '7'), it provides insight into how the model allocates uncertainty structurally, rather than showing per-sample predictive uncertainty.
>
> > The results given in Figure 4 seem dubious
>
> Thank you for your keen observation regarding Figure 4. We identified an issue in the plotting code of the Brier scores and have since corrected and replotted the results. The updated Brier score values are now consistent with the expected range reported in prior literature for MNIST. We appreciate the reviewer's comment in bringing this to our attention, which helped to improve the accuracy of the results. Brier score for Ensemble is the lowest for all models on ID data.
>
> > For Figure 5, I believe that plotting relative error is more apt for the appendix.... I more care about the absolute error. Also, the authors will need to provide the details of the subnetwork LLA method. The same issues arise for Figure 6
>
> Figures 5, 6: We have updated the plot to reflect absolute values for all metrics, Accuracy, Brier Score, and NLL. For the Linearized Laplace method, we retrieved the results from Daxberger et al [1] as per the caption, and refer the readers to the original publication for implementation details.

---

> > ### Author Response · Authors · 2025-07-10
> > **Response1-cont.**
> >
> > **Comments**
> >
> > > Further, the paper writes off post-hoc methods for subnetwork inference by stating of them: “this approach is limited as it relies on training a dense BNN as a first step and is followed by a post-hoc analysis for pruning “. Is this not exactly what the PBN method does?
> >
> > With regards to the clarity on the post-hoc methods, we would like to clarify the statement “this approach is limited as it relies on training a dense BNN as a first step and is followed by a post-hoc analysis for pruning,” refers to pruning strategies post-hoc to **Bayesian** Neural Network methods, which require first training a **dense Bayesian neural network** and then applying pruning strategies afterward. While PBN method is indeed post-hoc, it requires a **deterministic** pre-trained network, then training a subnetwork of parameters through variational inference. This distinction is significant, as PBN method does not require the costly training of a full Bayesian model upfront, thereby improving efficiency and practicality.
> >
> > > Comparison to other subspace uncertainty quantification methods needs to be looked at, such as the PCA method of [4] and [3], and how your subspace selection technique improves/detracts performance compared to theirs.
> >
> > We have presented comparative results for Daxberger et al. [1] (referred to as Linearized Laplace) in Figure 5. Our method outperforms Linearized Laplace under OOD datashift, demonstrating superior robustness. While reference [3] provides evaluations on regression benchmarks (UCI datasets) and in-distribution performance on CIFAR-10 and CIFAR-100, it does not include evaluations under distributional shift or OOD settings. As a result, a direct comparison to our method, which includes evaluation of robustness under distributional shift, is not feasible at this time.
> >
> > > There is a major issue with this paper: the similarity of this paper to [2] needs to be discussed.
> >
> > With regards to the stated overlap with reference [2]: While the current method builds upon concepts introduced in prior work [2], this submission makes distinct and substantive contributions that go well beyond that preliminary proof-of-concept study. Specifically, the earlier work focused on medical image classification and binary segmentation with relatively modest-scale datasets (e.g., ~110K images for classification and ~3.6K images for binary segmentation). In contrast, the current manuscript investigates the scalability and generalizability of the approach to significantly larger-scale computer vision tasks, including classification on ImageNet (1.2 million images) and multi-class segmentation on Cityscapes (20-class segmentation). Moreover, this submission offers a more comprehensive methodological description and discussion, along with extensive evaluations on robustness under distribution shift, which were not previously addressed.
> >
> >
> > **Requested Changed**
> >
> > 1. Brackets were added to citations in the revised manuscript.
> > 2.  On related work: We mention both SWAG and subnetwork LLA in the section that follows, on partial BNNs. We have included the missing mentioned methods to the computational demands of BNN section as per the suggestion. Thank you.
> >
> > > 3. The sensitivity-analysis section was under-whelming. This theory needs to be tightened to the core relationship...... Interestingly, this seems to be the inverse method to [3], where they take parameters with maximum marginal variance, as measured by the inverse of the GGN-Hessian, i.e. inverse of the gradient squared.
> >
> > 3. While the sensitivity analysis in Section 3 is concise, this simplicity is a strength of the proposed method. It effectively supports the subnetwork selection criterion and is grounded in standard uncertainty propagation through a function. We have elaborated on the method and its background with full derivations in Appendix J for the relationship of output uncertainty to sensitivity and individual parameter uncertainty.
> > This is a valid observation regarding the method for subnetwork selection in Daxberger, et al. [1]. We view this as a complementary approach to ours, both grounded in Equation 4 (section 3): $\left(\sigma^2_y\right) = \sum_{i=1}^{N}{\left(\frac{\partial f}{\partial \theta_i}\right)^2 \sigma^2_{\theta_i}}$ which shows that the output uncertainty depends on the sum of the product of parameter sensitivities and their variances. In our case, we assume a mean-field approximation and do not estimate parameter variances, so we rely on the sensitivity term as a selection criterion as an uncertainty importance measure. In contrast, Daxberger et al. [1] diagonalize the Hessian, effectively assuming independence, and select parameters based on their maximum marginal variances (the second term in equation 4). Both methods are valid and principled, with the choice driven by the available computational resources and modeling assumptions.

---

> > > ### Author Response · Authors · 2025-07-10
> > > **Response1-Cont.**
> > >
> > > **Requested Changes**
> > >
> > > > Method section needs a lot of work.
> > >
> > > The method section has been rehauled as per the requests. Experimental details for parameters such as the number of data points and the number of posterior samples are included in Appendix H. Here, $x, y$ refer to inputs and outputs, respectively. In the revised manuscript, notations have been revised to avoid confusion.
> > >
> > > > Notation of ρ,μ_posti,σ_posti,μ_{ij} was very confusing. What is i,j here? Also if you set ρ=0 for the deterministic parameters, won’t σ=ln⁡2?
> > >
> > > Deterministic $\rho_d$ values are set to a near-zero value for numerical stability; further, they are frozen during training, with gradients set to zero, so they do not contribute to the KL divergence term. (please see Sparse Gradient Updates section for more details).
> > >
> > > > Is Entropy of Expectation summed over all classes c?
> > >
> > > The entropy of expectation is averaged over all of the classes yes.
> > >
> > > > Claim of 50% and 80% reduction don’t make sense. Can’t you choose the number of parameters? At 1%, wouldn’t the reduction be 98%, as there are mean and variance for every ‘Bayesian’ parameter? Wouldn’t the ensemble reduction be simply 5x the reduction for a single network? You also claim at one point that reduction is > 95%.
> > >
> > > With regards to the relative reduction in number of parameters, when comparing to standard variational inference, the relative decrease in the number of *Bayesian* parameters is $>95\%$ (when $r_{bayes} <= 5\%$, but for the *total* number of trainable parameters, the relative decrease is >45\%. With respect to the total number of parameters for an Ensemble, if $r_{bayes} <= 5\%$ the relative decrease is $>=3.95\times$ or approximately $80\%$, but more precisely $79\%$. To avoid any confusion, we changed all references in relative parameter reduction to the total number of trainable parameters. For reference, in Appendix B we present a general form of number of parameters and FLOPs relative to a given deterministic model for comparative methods. This has been clarified in the revised manuscript.
> > >
> > > **References:**
> > > 1. Daxberger, et al (2021). ICML. Bayesian Deep Learning via Subnetwork Inference.
> > > 2. Abboud, Lombaert, Kadoury. Sparse bayesian networks: Efficient uncertainty quantification in medical image analysis.
> > > 3. Izmailov, Pavel, et al. (2019). UAI. Subspace inference for Bayesian deep learning.

---

### Review · Reviewer_aVs3 · 2025-06-09

**Summary Of Contributions:**

The authors introduced an algorithm for training scalable partial Bayesian networks that starts from a pre-trained deterministic model and selectively assigns a small portion of Bayesian parameters based on first-order gradient analysis.

A wide range of experiments analyzed the proposed method and demonstrated its effectiveness in estimating predictive uncertainty while being computationally efficient.

**Audience:**

Yes

**Broader Impact Concerns:**

No concerns on the ethical implications, from my perspective.

**Claims And Evidence:**

Yes

**Requested Changes:**

Please refer to the comments on *Weaknesses and Questions* and *Minor comments*. Thanks!

**Strengths And Weaknesses:**

**Strengths**
1. The paper is organized in a clear structure, and the motivation is clearly stated.

2. The paper is relatively easy to follow, and the literature study is ok.

3. A wide range of experimental validations were performed, considering different tasks, various datasets and neural network architectures, as well as distinct hyperparameter settings, among others.

**Weaknesses and Questions**
1.  In this work, the uncertainty estimation of the proposed method uses the posterior sampling via multiple forward passes ($N$ samples). It seems that the $N$ used for the proposed method and the baselines is not mentioned in the main body. Additionally, I would suggest adding an introduction (relative equations) about how such Bayesian models estimate uncertainties. Further, I would suppose that the uncertainty evaluation is based on the final (averaged) prediction, i.e., total uncertainty. The following question is, will the epistemic uncertainty estimation quality of the proposed method be comparable to that of VI and deep ensembles?

2. Regarding the evaluation on uncertainty quantification, the OOD detection task is a widely applied quantitative proxy in image classification tasks.  Adding those comparisons will highlight the contribution of the proposed method.

3. Given a set of well-trained individual neural networks within a deep ensemble, the results of first-order gradient-based sensitivity analysis are likely to differ across the ensemble members. In this context, could the authors clarify whether they have specific criteria or insights guiding the selection of a particular network for constructing the proposed PBN?

**Minor comments**

a. Regarding the 'contribution claim' on page 2, it will be clearer to include the number of ensembles concerning the comparison.

b. Will it be possible to add a figure to illustrate the initialization process of the PBN (partially concerning the mask parts) to enhance the clarity? (Just recommend)

c. Placing the result figures after (or close to) the relative text will enhance the clarity.

d. Detailed results in Figures 4 (c) and (d) are not highly readable. Maybe consider increasing the figure size.

---

> ### Author Response · Authors · 2025-07-10
>
> We thank the reviewer for their comments and feedback for improving the manuscript. Please find our responses to the questions/Comments:
>
> **Clarifications:**
> 1. >In this work, the uncertainty estimation of the proposed method uses the posterior sampling via multiple forward passes (
>  samples). It seems that the  used for the proposed method and the baselines is not mentioned in the main body. Additionally, I would suggest adding an introduction (relative equations) about how such Bayesian models estimate uncertainties. Further, I would suppose that the uncertainty evaluation is based on the final (averaged) prediction, i.e., total uncertainty. The following question is, will the epistemic uncertainty estimation quality of the proposed method be comparable to that of VI and deep ensembles?
>     - The number of N samples is 5, and the number of ensemble members is also 5. The description of the training details is in Appendix H, with details on the variational inference training in Appendix H.6.
>     -  Regarding relative equations for uncertainty estimation through Bayesian models - we updated Section 3 to reflect uncertainty estimation through variational inference.
>     - To answer the reviewer's question, yes, the uncertainty is computed based on averaged multiple predictions based on the N forward passes.
>     - Based on our experiments, the uncertainty estimated via PBN is better than that from VI and comparable to that of deep ensembles.
> 2. >Regarding the evaluation on uncertainty quantification, the OOD detection task is a widely applied quantitative proxy in image classification tasks. Adding those comparisons will highlight the contribution of the proposed method.
>     - In addition to the uncertainty-based OOD analysis, we have included an AUROC plot for OOD detection between MNIST and FashionMNIST in Figure 4(d). This plot compares various partial networks with baseline methods (Ensemble, VI, Deterministic), showing that Partial 10\% achieves the best OOD detection performance, on par with the Ensemble.
> 3. >Given a set of well-trained individual neural networks within a deep ensemble, the results of first-order gradient-based sensitivity analysis are likely to differ across the ensemble members. In this context, could the authors clarify whether they have specific criteria or insights guiding the selection of a particular network for constructing the proposed PBN?
>     - Thank you for bringing this point regarding the selection of the network. In our experiments, we assume that the pre-trained deterministic network with the best performance on the validation set can be used as an initialization for the PBN network. It is worth noting that the individual parameters within each layer might vary across the different networks; however, based on our experiments, the selected subnetwork is limited to the first and last few layers.
>
>
> 1. Minor comments:
>     - Number of ensembles is included in Appendix H.
>     - Yes, we have added an illustration to the initialization process in Appendix I.
>     - Figure order with text has been updated.
>     - Figure 4 has been updated with clearer plots for better legibility.

---

### Review · Reviewer_Kbbt · 2025-06-18

**Summary Of Contributions:**

This work presents a way to convert the deterministic DNNs into partial BNNs and their training. Specifically, for the deterministic DNNs trained by MAP, the proposed method selects the partial layers to be Bayesian based on the magnitude of the gradient (larger chosen) because the magnitude of the gradient can represent the sensitivity of the model output based on first-order Taylor approximation. Then, the proposed method regards the parameters of chosen layers as random variables, and it then trains the parameters of their variational distributions using Variational inference (VI) with the reparameterization trick (RP) while training the left parameters of unchosen layers using MAP. The proposed method has been extensively demonstrated in toy regression, image classification, and pixel-label segmentation.

**Audience:**

Yes

**Claims And Evidence:**

No

**Requested Changes:**

## Requested Changes

### 1. Improve writing in Section 2 and 3.
> I ask the authors to elaborate the explanation for Eq. (2) and (3). In addition, In current form, it is written for the deterministic parameters. Please explain why the deterministic parameter with large sensitivity should be modeled as random variable.

> Partial Bayesian Network --> Partial Bayesian Neural Network; Bayesian network and  Bayesian Neural Network are different.

> "This approach ensures a robust starting point for optimization uncertainty learning in the partial Bayesian network" is repeated in section 3.

> $\mathcal{N}_i(\mu_i,0) = \delta_i(\mu_i)$ is not correct because  Gaussian distribution with zero variance is deterministic, not a random variable. What you claim is that  $\mathcal{N}_i(\mu_i, \sigma^2)$ converges to the Dirac delta $\delta_i(\mu_i)$ in the limit sense as $\sigma^2 \to 0$, i.e., in the asymptotic sense of weak convergence of measures.



### 2. Explain why the empirical results of DE is different to that of [1] in Section 4.

> I recommend to investigate whether the experiment setting for DE follows the same protocol of [1]


### 3. Strengthen the explanation of why the proposed method can yield better results than Linearized Laplace.
> Also, it would also be helpful to explain the computational efficiency of the proposed method compared to Linearized Laplace. I believe that such a comparison would strengthen the manuscript.


## Questions.
*  I am curios about  why the zero-mean prior is used when the pre-trained parameters are used. Have you considered the non-zero mean prior where the mean is set as the values of the pre-trained parameters ?

* How many ensemble members are used in the Deep Ensemble (DE) results shown in Figures 5 and 6 ?


[1] Can You Trust Your Model’s Uncertainty? Evaluating Predictive Uncertainty Under Dataset Shift - NeurIPS 19

**Strengths And Weaknesses:**

### Strengths

* The proposed method is simple and appears easy to use with deterministic pre-trained models, which are commonly employed in practice.

* The method demonstrates superior performance across various experiments.

### Weaknesses

* Less clear on performance improvement.
> The magnitude of the gradient for the parameter could be a reasonable criterion for choosing the partial layers to be a Bayesian. However, other approaches such as posterior approximation [1] and signal-to-ratio [2]  can be also considerable  for sub-network selection.  While I appreciate that this simple method can produce strong partial BNNs from deterministic DNNs, the manuscript currently lacks a clear explanation for why the proposed selection method leads to superior performance compared to alternative strategies.



* Less credible on empirical results on Deep ensemble (DE).
> The DE is known to outperform other BNNs in general including corrupted CIFAR-10 and ImageNet, which is consistent with my empirical experiences; please check that the DE outperforms other baselines including Dropout, as shown in Figure 2 and Appendix C in [3] in terms of ACC, ECE, and NLL.
However, following Figure 5 and Figure 6,  DE is extremely worse than Dropout. I feel hard to agree with this result and remain skeptical about the claimed performance improvements of your method.

* Writing of some part are not clear.
> For example, in Eq.(2) and (3), if $\Theta$ denote the random variable, it should be written in sense of Expectation. Otherwise, the mean-field assumption, where each random variable of variational posterior distribution assumed to be independent, does not seem valid. Also, some sentences are repeated in **Initialization of Partial Bayesian Network** in Section 3.



[1] Bayesian Deep Learning via Subnetwork Inference - ICML 21

[2] Training Bayesian Neural Networks with Sparse Subspace Variational Inference - ICLR 24

[3] Can You Trust Your Model’s Uncertainty? Evaluating Predictive Uncertainty Under Dataset Shift - NeurIPS 19

---

> ### Author Response · Authors · 2025-07-10
>
> We thank the reviewer for their comments and feedback for improving the manuscript. Please find our responses to the questions and comments:
>
> **Comments**
> >  1. the manuscript currently lacks a clear explanation for why the proposed selection method leads to superior performance compared to alternative strategies.
>
> The focus of the manuscript is on how to integrate partial Bayesian layers effectively and empirically demonstrate the scalability of the gradient-magnitude-based selection, which yields competitive performance. A formal investigation into *why* this criterion outperforms other methods lies beyond the scope of this conference paper. We view it as an important direction for future work.
>
> > 2. Less credible on empirical results on Deep ensemble (DE).
>
> For CIFAR-10, the data for Linearized Laplace, ensemble, and dropout were originally taken from Daxberger, et al (2021) [1], and the first version of the paper included the *relative* performance on corrupted data versus that of the uncorrupted data $\Delta (m) = | m_{c_i} - m_{test} | / m_{test}$ where $m$ is a metric, $c_i$ is corruption at intensity $i$ and $m_{test}$ is performance for test dataset without corruption. We have updated the plots to reflect the *actual* metrics (not relative) to avoid confusion. And have included the data from Ovadia (2019) [2] for both figures 5, 6 except for the data related to Linearlized Laplace which is retrieved from Daxberger (2021) [1].
>
> > 3. Writing of some part are not clear.
>
>  Clarity for Section 3 has been improved. Please find the revision in the updated manuscript.
>
> **Requested Changes**
>
> >  Improve writing in Section 2 and 3. Please explain why the deterministic parameter with large sensitivity should be modeled as random variable.
>
> We have improved the quality of writing in sections 2 and 3 in the latest revision, and have elaborated in Section 3 on the sensitivity analysis.
>
> Equations 2 and 3 are based on standard uncertainty propagation, on the impact of uncertainty in individual variables, and on the uncertainty in the output of a function parametrized by those variables. We have updated the derivation in section 3 and included the full derivation of the relationship between output uncertainty, individual parameter uncertainty, and sensitivity in Appendix J.
>
> The equations are formulated using deterministic parameters to provide a basis for selecting which parameters to treat as stochastic during the variational inference training phase. As illuastrated in Equation 4, the contribution of each parameter’s uncertainty $(\sigma^2_{\theta_i})$ to the output uncertainty $(\sigma^2_y$  is scaled by the sensitivity term $\left(\frac{\partial f}{\partial \theta_i}\right)$. Since the goal is to estimate output uncertainty $\Delta y$, which depends on the unknown parameter uncertainties $(\sigma^2_{\theta_i})$, the only available indicator of each parameter’s influence on output uncertainty is the sensitivity term. The first part of the new revision of Section 3 might provide better insight into the discussion.
>
> Regarding the explanation/notation: "Deterministic parameters are modeled as $\mathcal{N}(\mu_{i}, 0) = \delta_{i}(\mu_{i})$ delta functions, while Bayesian parameters are modeled as $\mathcal{N}(\mu_{i}, \sigma_{i})$ distributions."
>
> This sentence refers to how the layers are *implemented* programmatically, to have a mixed deterministic and Bayesian parameters in *individual layers*. This is explained in detail in the last paragraph in the "Initialization of Partial Bayesian Neural Network" section. To avoid confusion, we rewrote the sentence as follows: "Deterministic parameters are modeled as $\delta(\mu_{i})$ delta functions, while Bayesian parameters are modeled as $\mathcal{N}(\mu_i, \sigma_i)$".
>
> >  Explain why the empirical results of DE is different to that of [1] in Section 4.
>
> Regarding deep ensemble results: clarification provided above in comments No.2.
>
> > Strengthen the explanation of why the proposed method can yield better results than Linearized Laplace.
>
>  The Linearized Laplace approximation models the posterior as a local Gaussian centered around a fixed MAP estimate, relying on a second-order approximation. In contrast, the partial variational inference framework maintains a flexible variational posterior over a subset of selected parameters, updating them jointly in the second training phase, and thereby allowing adaptation to more complex or non-local posterior structures that may arise during training.

---

> > ### Author Response · Authors · 2025-07-10
> >
> > **Questions**
> >
> > > 1. Why the zero-mean prior is used when the pre-trained parameters are used? Have you considered the non-zero mean prior where the mean is set as the values of the pre-trained parameters ?
> >
> > 1) An isotropic zero-mean Gaussian acts as a regularizer, analogous to weight decay in standard deterministic training.
> > 2) It simplifies the computation of the KL divergence, leading to a closed-form expression of the KL.
> > 3) It follows standard practice in variational inference [3-6].
> >
> >     Regarding the suggestion to set individual priors based on the pretrained parameter values, this is fundamentally different from the PBN approach. We assume a single prior across all parameters to act as a regularizer,  hence the standard isotropic Gaussian. Using parameter-specific priors centered at the pretrained weights would introduce additional computational complexity, as one would need to store separate prior means and compute individualized KL terms, complicating the reparameterization trick. The subnetwork PBN approach does not constrain the mean of the variational posterior to remain fixed at the pretrained weights. Rather, we initialize the posterior means with the pretrained values and then allow them to be tuned during the second-stage training to optimize the ELBO objective. This provides a flexible adaptation to the new data while still benefiting from the pretrained initialization.
> >
> > > 2. How many ensemble members are used in the Deep Ensemble (DE) results shown in Figures 5 and 6 ?
> >
> > The data in Figure 5 (CIFAR10) and 6 (ImageNet) for comparative performance were retrieved from [1] (for Linearized Laplace), [2] (remaining baselines), as mentioned in the captions. For CIFAR10/ImageNet, a 10-member ensemble was used as per [2].
> >
> > **References:**
> > 1. Daxberger, et al (2021). ICML. Bayesian Deep Learning via Subnetwork Inference.
> > 2. Ovadia, et al (2019). NeurIPS.  Can you trust your model’s uncertainty? Evaluating predictive uncertainty under dataset shift.
> > 3. Kingma and Welling (2013) ICLR. Auto-Encoding Variational Bayes.
> > 4. Harrizon, Willes, and Snoek (2024) ICLR. Variational Bayesian Last Layers.
> > 5. Graves (2011) NeurIPS. Practical Variational Inference for Neural Networks.
> > 6. Sharma, et al. (2023) AISTATS. Do Bayesian Neural Networks Need To Be Fully Stochastic?

---

### Decision · Action_Editor_yrkY · 2025-07-31

**Recommendation:** Reject

**Audience:**

Yes

**Audience Explanation:**

No reviewers raised objections regarding the paper’s alignment with the second criterion related to "Audience."

**Claims And Evidence:**

No

**Claims Explanation:**

While the proposed method offers an intuitive approach to converting pre-trained models into partial BNNs, the empirical evidence and experimental design fall short of convincingly demonstrating its benefits. Both reviewers note issues with experimental clarity, weak comparative analysis, and inconsistent or marginal performance gains over baselines. Substantial revision and more rigorous experimentation are needed to support the paper’s claims.